# LRRK2 maintains mitochondrial homeostasis and regulates innate immune responses to *Mycobacterium tuberculosis*

Chi G Weindel[1†], Samantha L Bell[1†], Krystal J Vail[2], Kelsi O West[1], Kristin L Patrick[1], Robert O Watson[1]*

[1]Department of Microbial Pathogenesis and Immunology, Texas A&M Health Science Center, Bryan, United States; [2]Department of Veterinary Pathobiology, Texas A&M University College of Veterinary Medicine and Biomedical Sciences, College Station, United States

**Abstract** The Parkinson's disease (PD)-associated gene leucine-rich repeat kinase 2 (*LRRK2*) has been studied extensively in the brain. However, several studies have established that mutations in *LRRK2* confer susceptibility to mycobacterial infection, suggesting LRRK2 also controls immunity. We demonstrate that loss of LRRK2 in macrophages induces elevated basal levels of type I interferon (IFN) and interferon stimulated genes (ISGs) and causes blunted interferon responses to mycobacterial pathogens and cytosolic nucleic acid agonists. Altered innate immune gene expression in *Lrrk2* knockout (KO) macrophages is driven by a combination of mitochondrial stresses, including oxidative stress from low levels of purine metabolites and DRP1-dependent mitochondrial fragmentation. Together, these defects promote mtDNA leakage into the cytosol and chronic cGAS engagement. While *Lrrk2* KO mice can control *Mycobacterium tuberculosis* (Mtb) replication, they have exacerbated inflammation and lower ISG expression in the lungs. These results demonstrate previously unappreciated consequences of LRRK2-dependent mitochondrial defects in controlling innate immune outcomes.

*For correspondence:
robert.watson@tamu.edu

†These authors contributed equally to this work

Competing interests: The authors declare that no competing interests exist.

## Introduction

Mutations in leucine-rich repeat kinase 2 (*LRRK2*) are a major cause of familial and sporadic Parkinson's disease (PD), a neurodegenerative disease characterized by selective loss of dopaminergic neurons in the substantia nigra pars compacta region of the midbrain (*Cookson, 2017*; *Kim and Alcalay, 2017*; *Martin et al., 2014*; *Schulz et al., 2016*). Despite LRRK2 having been implicated in a variety of cellular processes, including cytoskeletal dynamics (*Civiero et al., 2018*; *Kett et al., 2012*; *Pellegrini et al., 2017*), vesicular trafficking (*Herbst and Gutierrez, 2019*; *Sanna et al., 2012*; *Shi et al., 2017*), calcium signaling (*Bedford et al., 2016*; *Calì et al., 2014*), and mitochondrial function (*Ryan et al., 2015*; *Singh et al., 2019*; *Yue et al., 2015*), its precise mechanistic contributions to triggering and/or exacerbating PD and other disease pathologies are not known.

Of all the cellular pathways affected by *LRRK2* mutations, dysregulation of mitochondrial homeostasis has emerged as a centrally important mechanism underlying PD pathogenesis and neuronal loss (*Cowan et al., 2019*; *Panchal and Tiwari, 2019*). Indeed, other PD-associated genes, such as *PARK2* (Parkin), *PINK1*, and *DJ1*, all play crucial roles in mitochondrial quality control via mitophagy. LRRK2 has been implicated in mitophagy directly through interactions with the mitochondrial outer membrane protein MIRO (*Hsieh et al., 2016*), and several lines of evidence support roles for LRRK2 in controlling mitochondrial network dynamics through interactions with the mitochondrial fission protein DRP1 (*Wang et al., 2012*). Accordingly, a number of different cell types, including fibroblasts and iPSC-derived neurons from PD patients harboring mutations in *LRRK2* exhibit defects in

**eLife digest** Parkinson's disease is a progressive nervous system disorder that causes tremors, slow movements, and stiff and inflexible muscles. The symptoms are caused by the loss of cells known as neurons in a specific part of the brain that helps to regulate how the body moves.

Researchers have identified mutations in several genes that are associated with an increased risk of developing Parkinson's. The most common of these mutations occur in a gene called *LRRK2*. This gene produces a protein that has been shown to be important for maintaining cellular compartments known as mitochondria, which play a crucial role in generating energy. It remains unclear how these mutations lead to the death of neurons.

Mutations in *LRRK2* have also been shown to make individuals more susceptible to bacterial infections, suggesting that the protein that *LRRK2* codes for may help our immune system. Weindel, Bell et al. set out to understand how this protein works in immune cells called macrophages, which 'eat' invading bacteria and produce type I interferons, molecules that promote immune responses. Mouse cells were used to measure the ability of normal macrophages and macrophages that lack the mouse equivalent to *LRRK2* (referred to as *Lrrk2* knockout macrophages) to make type I interferons.

The experiments showed that the *Lrrk2* knockout macrophages made type I interferons even when they were not infected with bacteria, suggesting they are subject to stress that triggers immune responses. It was possible to correct the behavior of the *Lrrk2* knockout macrophages by repairing their mitochondria. When mice missing the gene equivalent to *LRRK2* were infected with the bacterium that causes tuberculosis, they experienced more severe disease.

The protein encoded by the *LRRK2* gene is considered a potential target for therapies to treat Parkinson's disease, and several drugs that inhibit this protein are being tested in clinical trials. The findings of Weindel, Bell et al. suggest that these drugs may have unintended negative effects on a patient's ability to fight infection. This work also indicates that *LRRK2* mutations may disrupt immune responses in the brain, where macrophage-like cells called microglia play a crucial role in maintaining healthy neurons. Future studies that examine how mutations in *LRRK2* affect microglia may help us understand how Parkinson's disease develops.

mitochondrial network integrity as well as increased reactive oxygen species (ROS) and oxidative stress (*Sison et al., 2018*; *Smith et al., 2016*). In spite of these well-appreciated links, LRRK2's contribution to mitochondrial health in cells outside of the brain remains vastly understudied.

There is mounting evidence that mutations in *LRRK2*, as well as in other genes related to PD including *PARK2* and *PINK1*, contribute to immune outcomes both in the brain and in the periphery. For example, mutations in *LRRK2* impair NF-κB signaling pathways in iPSC-derived neurons and render rats prone to progressive neuroinflammation in response to peripheral innate immune triggers (*López de Maturana et al., 2016*), and chemical inhibition of LRRK2 attenuates inflammatory responses in microglia ex vivo (*Moehle et al., 2012*). In addition to these strong connections between *LRRK2* and inflammatory responses in the brain, numerous genome-wide association studies suggest that LRRK2 is an equally important player in peripheral immune responses. Single nucleotide polymorphisms (SNPs) in *LRRK2* are associated with susceptibility to mycobacterial infection, inflammatory colitis (*Umeno et al., 2011*), and Crohn's disease (*Van Limbergen et al., 2009*). Consistent with a role for LRRK2 in pathogen defense and autoimmunity, it is abundant in many immune cells (e.g. B cells, dendritic cells, monocytes, macrophages), and expression of *LRRK2* is induced in human macrophages treated with IFN-γ (*Gardet et al., 2010*). Loss of LRRK2 reduces IL-1β secretion in response to *Salmonella enterica* infection in macrophages (*Liu et al., 2017*) and enhances expression of pro-inflammatory cytokines in response to *Mycobacterium tuberculosis* (Mtb) infection at early time points of mouse infection (*Härtlova et al., 2018*). However, the precise mechanistic contributions of LRRK2 to controlling immune responses in the periphery remain poorly understood.

Here, we provide evidence that LRRK2's ability to influence inflammatory gene expression in macrophages is directly linked to its role in maintaining mitochondrial homeostasis. Specifically, we demonstrate that mitochondrial stress and hyper-activation of DRP1 in *Lrrk2* KO macrophages leads to

the release of mitochondrial DNA (mtDNA), chronic engagement of the cGAS-dependent DNA sensing pathway, and abnormally elevated basal levels of type I IFN and ISGs. These high basal levels of type I IFN appear to completely reprogram *Lrrk2* KO macrophages, rendering them refractory to a number of distinct innate immune stimuli, including infection with Mtb. While Mtb-infected *Lrrk2* KO mice did not exhibit significant differences in bacterial burdens compared to controls, we did observe exacerbated pathology and lower expression of ISGs in the lungs at early infection timepoints. Collectively, these results demonstrate that LRRK2's role in maintaining mitochondrial homeostasis is critical for proper induction of type I IFN gene expression in macrophages and for downstream inflammatory responses during in vivo infection.

## Results

### RNA-seq analysis reveals that LRRK2-deficiency in macrophages results in dysregulation of the type I IFN response during Mtb infection

To begin to implicate LRRK2 in the peripheral immune response, we took an unbiased approach and asked how loss of LRRK2 impacts innate immune gene expression during Mtb infection of macrophages ex vivo. Briefly, primary murine bone marrow-derived macrophages (BMDMs) derived from littermate heterozygous (HET) and knockout (KO) *Lrrk2* mice were infected with Mtb at MOI of 10. RNA-seq analysis was performed on total RNA collected from uninfected and infected cells 4 hr post-infection (*Lrrk2* KO n = 4, *Lrrk2* HET n = 3). Previous studies have identified 4 hr as a key innate immune time point during Mtb infection, corresponding to the peak of transcriptional activation downstream of several pattern recognition receptors (PRRs), including the cytosolic DNA sensor cGAS (*Manzanillo et al., 2012*; *Watson et al., 2015*; *Watson et al., 2012*).

Following analysis with CLC Genomics Workbench, we first asked whether we could detect gene expression differences in uninfected *Lrrk2* HET and KO macrophages. Surprisingly, we identified hundreds of genes whose expression was significantly higher in *Lrrk2* KO macrophages (blue genes, *Figure 1A*). Taking a closer look at the most affected genes (zoom-in, right), we noted that a number of well-characterized ISGs (e.g. *Mx1*, *Ifit1*, *Irf7*, *Rsad2*, etc.) were expressed several times higher in macrophages lackingLRRK2 (p<0.05). These trends persisted when we compared *Lrrk2* WT vs. KO or *Lrrk2* HET vs. KO (*Figure 1—figure supplement 1B-C*). Unbiased canonical pathway analysis confirmed a global upregulation of ISGs, identifying 'Interferon signaling' and 'Activation of IRF by cytosolic PRRs' as the top enriched pathways in uninfected *Lrrk2* KO vs. HET BMDMs (*Figure 1B*).

We next looked at gene expression differences in *Lrrk2* KO vs. HET BMDMs at 4 hr post-infection with Mtb. Mtb is a potent activator of type I IFN expression, thought to occur mostly through perturbation of the Mtb-containing phagosome and release of bacterial dsDNA into the cytosol, where it is detected by DNA sensors like cGAS, activating the STING/TBK1/IRF3 axis (*Collins et al., 2015*; *Wassermann et al., 2015*; *Watson et al., 2015*; *Wiens and Ernst, 2016*). Curiously, many of the same ISGs whose expression was statistically higher at baseline in *Lrrk2* KO BMDMs failed to induce to the same levels following Mtb infection (e.g. *Ifit*, *Cmpk2*, *Gbp2*, *Rsad2*; *Figure 1C*, orange genes, zoom-in, left). This blunted global type I IFN response was also evident via qualitative assessment of genes whose expression was measurably lower in Mtb-infected *Lrrk2* KO BMDMs but failed to demonstrate statistical significance (*Figure 1—figure supplement 1A*). Again, canonical pathway analysis identified an enrichment for immune genes whose expression was impacted by loss of LRRK2 in response to Mtb (*Figure 1D*). RT-qPCR analysis confirmed higher baseline expression and lower induction during Mtb infection of several ISGs: *Rsad2*, *Gbp2*, *Cmpk2*, *Stat2*, and *Ifit1* in *Lrrk2* KO BMDMs (*Figure 1E*; see 'Statistical analysis' section in Materials and methods for details regarding the statistical analysis of baseline and induced gene expression). We also measured high basal levels of *Ifnb* and *Isg15*, although the differences in induction of these genes between *Lrrk2* KO and HET macrophages were more modest in this particular experiment (*Figure 1E*). Increased basal expression and decreased induction of IFN and ISGs was also detected during Mtb infection in the human monocyte cell line U937 (*Figure 1—figure supplement 1D*) and in RAW 264.7 murine macrophages when *Lrrk2* expression was knocked down by shRNA (*Lrrk2* KD) (*Figure 1—figure supplement 1E*). Importantly, blunted expression was not observed for all immune genes; for example, loss of *Lrrk2* had no effect on the NFκB gene *Tnfa* despite the transcript being dramatically induced upon Mtb infection (*Figure 1F*). Interestingly,

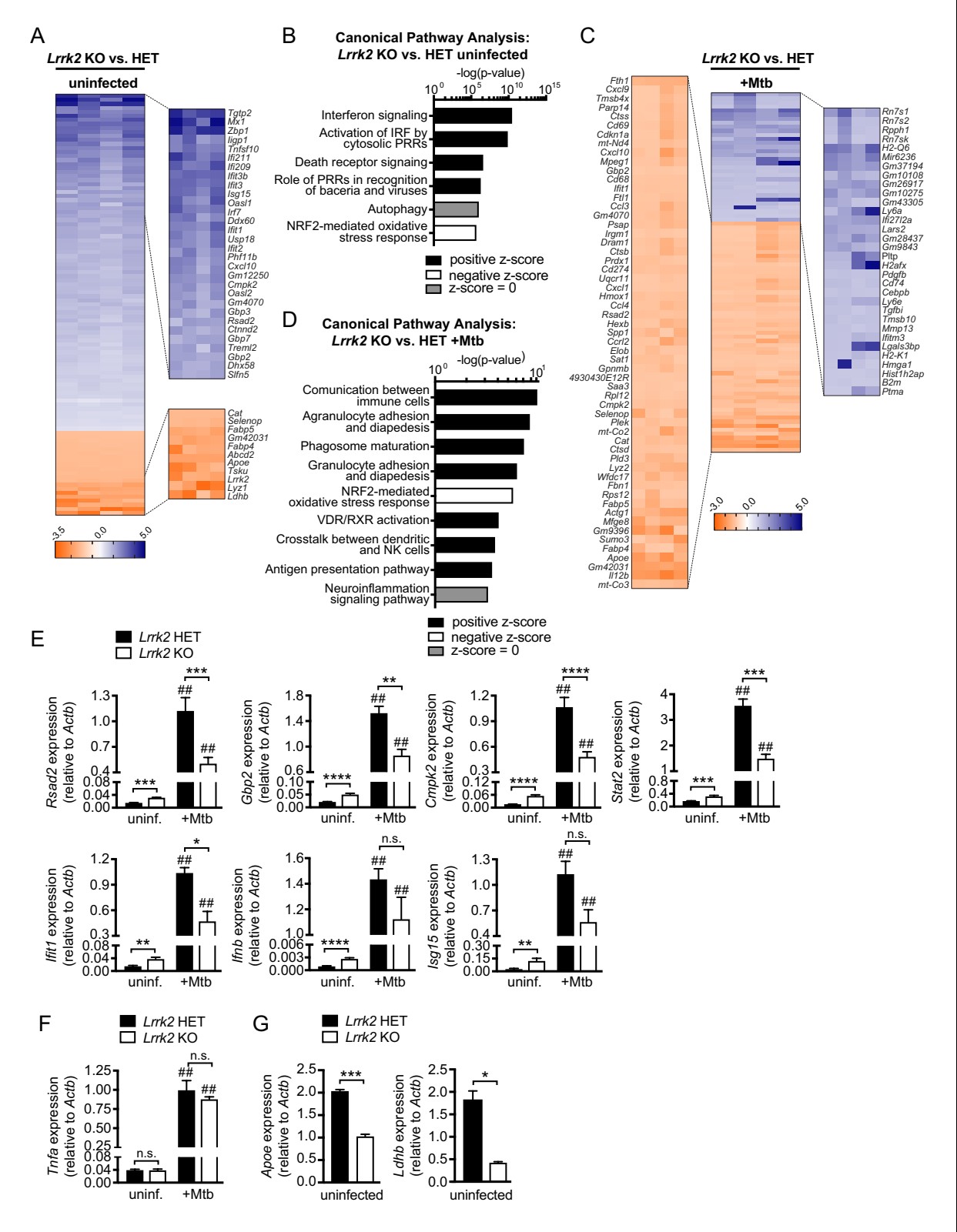

**Figure 1.** Global gene expression analysis reveals that *Lrrk2* KO macrophages are deficient at inducing type I IFN expression and have higher basal levels of ISGs. (**A**) Heatmap depicting significant gene expression differences (Log2 fold-change, p<0.05) between uninfected *Lrrk2* KO and HET BMDMs. (**B**) IPA software analysis showing cellular pathways enriched for differentially expressed genes in uninfected *Lrrk2* KO vs. HET BMDMs. (**C**) Heatmap depicting significant gene expression differences (Log2 fold-change) between *Lrrk2* KO and HET BMDMs during infection with Mtb. (**D**) As in

*Figure 1 continued on next page*

Figure 1 continued

(B) but for pathways enriched for differentially expressed genes in Mtb-infected *Lrrk2* KO and HET BMDMs, 4 hr post-infection. (E) RT-qPCR showing expression of *Ifnb* and IFN stimulated genes in uninfected and Mtb-infected *Lrrk2* KO and HET macrophages. Data are shown as *ISG/Actb*. (F) RT-qPCR of *Tnfa* in *Lrrk2* KO and HET BMDMs. (G) RT-qPCR of *Apoe* and *Ldhb* normalized to *Actb* in uninfected BMDMs. Throughout the manuscript, data are expressed as a mean of three or more biological replicates with error bars depicting SEM. Statistical tests used can be found at the end of the legend. Statistical analysis: *p<0.05, **p<0.01, ***p<0.005, ****p<0.001 (comparing indicated data points); ##p<0.001 (comparing stimulated to unstimulated of same genotype). In (E–F) a two-way ANOVA Tukey post-test was applied, and in (G) a two-tailed Student's T test.

The online version of this article includes the following figure supplement(s) for figure 1:

**Figure supplement 1.** Lrrk2 KO macrophages fail to induce proper levels of ISG expression in response to Mtb infection.

expression of several non-ISG, non-immune genes was reduced in uninfected *Lrrk2* KO BMDMs, including *ApoE*, which has been repeatedly linked to inflammatory and neurodegenerative diseases, and *Ldhb*, a critical metabolic gene involved in post-glycolytic energy production (*Figure 1G*). Collectively, these transcriptome-focused analyses revealed that *Lrrk2* KO macrophages have a high baseline IFN signature but generally fail to induce the type I IFN response to the same level as control cells when infected with Mtb. This phenotype is unusual and suggests that *Lrrk2* KO macrophages are somehow fundamentally reprogrammed. Typically, high resting IFN levels potentiate type I IFN responses, leading to a hyperinduction of ISGs following innate immune stimuli (*West et al., 2015*; *Yang et al., 2018*).

## *Lrrk2* KO macrophages exhibit blunted type I IFN induction in response to cytosolic nucleic acid agonists

We next wanted to define the nature of the innate immune stimuli that would elicit a blunted type I IFN response in *Lrrk2* KO macrophages. We began by infecting macrophages with *Mycobacterium leprae* (Mlep), which shares a virulence-associated ESX-1 secretion system with Mtb and also induces type I IFN through cytosolic nucleic acid sensing (*de Toledo-Pinto et al., 2016*). We measured a significant defect in ISG expression 8 hr post-infection in *Lrrk2* KO BMDMs and *Lrrk2* KO RAW 264.7 macrophages compared to control cells (*Figure 2A* and *Figure 2—figure supplement 1A*). We next treated primary macrophages and macrophage cell lines with a panel of agonists designed to elicit type I IFN expression downstream of a variety of PRRs. Transfection of immunostimulatory dsDNA (ISD), which is recognized by cGAS and stimulates the STING/TBK1/IRF3 axis, induced blunted *Ifnb* expression in *Lrrk2* KO BMDMs (*Figure 2B*), *Lrrk2* KO peritoneal macrophages (PEM) (significant differences in *Ifnb* expression were measured between *Lrrk2* KO and HET at baseline but induction differences failed to reach statistical significance via 2-way ANOVA Tukey post-hoc testing) (*Figure 2C*), *Lrrk2* KO RAW 264.7 macrophages (*Figure 2D*), *Lrrk2* KO mouse embryonic fibroblasts (MEFs) (*Figure 2E*), and *Lrrk2* KD RAW 264.7 macrophages (*Figure 2F*). Consistent with the BMDM phenotype from *Figure 1*, we observed higher basal expression of *Ifnb*/ISGs and a blunted response to ISD in all the *Lrrk2* KO/KD cell lines tested (*Figure 2A–E* and *Figure 2—figure supplement 1C*). We also found that *Lrrk2* KO BMDMs failed to fully induce *Ifnb* if we bypassed cGAS and stimulated the DNA sensing adapter STING directly using the agonist DMXAA (*Figure 2G*). In support of a defect in cytosolic nucleic acid sensing and IFNAR signaling, western blot analysis of IRF3 (phospho-Ser396) and STAT1 (phospho-Tyr701) activation showed a significant defect in the ability of *Lrrk2* KO macrophages to signal through IFNAR (phospho-STAT1) and a modest defect in cytosolic DNA sensing (phospho-IRF3) over the course of 6 hr following ISD transfection (*Figure 2H*, quantitation below, and *Figure 2—figure supplement 1B*). Collectively, these results suggest that type I IFN-generating pathways are chronically activated in cells lacking LRRK2, but their induction is muted compared to controls when faced with agonists of the cytosolic DNA sensing pathway.

We next tested whether loss of LRRK2 impacted the ability of cells to respond to activators of the type I IFN response outside of the cytosolic DNA sensing cascade. To this end, we treated *Lrrk2* KO and HET BMDMs with transfected poly(I:C) (to activate cytosolic RNA sensing), LPS (to stimulate TRIF/IRF3 downstream of TLR4), and CpG and CL097 (to stimulate nucleic acid sensing via TLR9 and TLR7, respectively). Interestingly, while we observed a defect in *Ifnb* induction in *Lrrk2* KO BMDMs stimulated with poly(I:C), we saw no difference in the ability of *Lrrk2* KO BMDMs to express type I IFNs following treatment with LPS, CL097, or CpG (*Figure 2I* and *Figure 2—figure supplement*

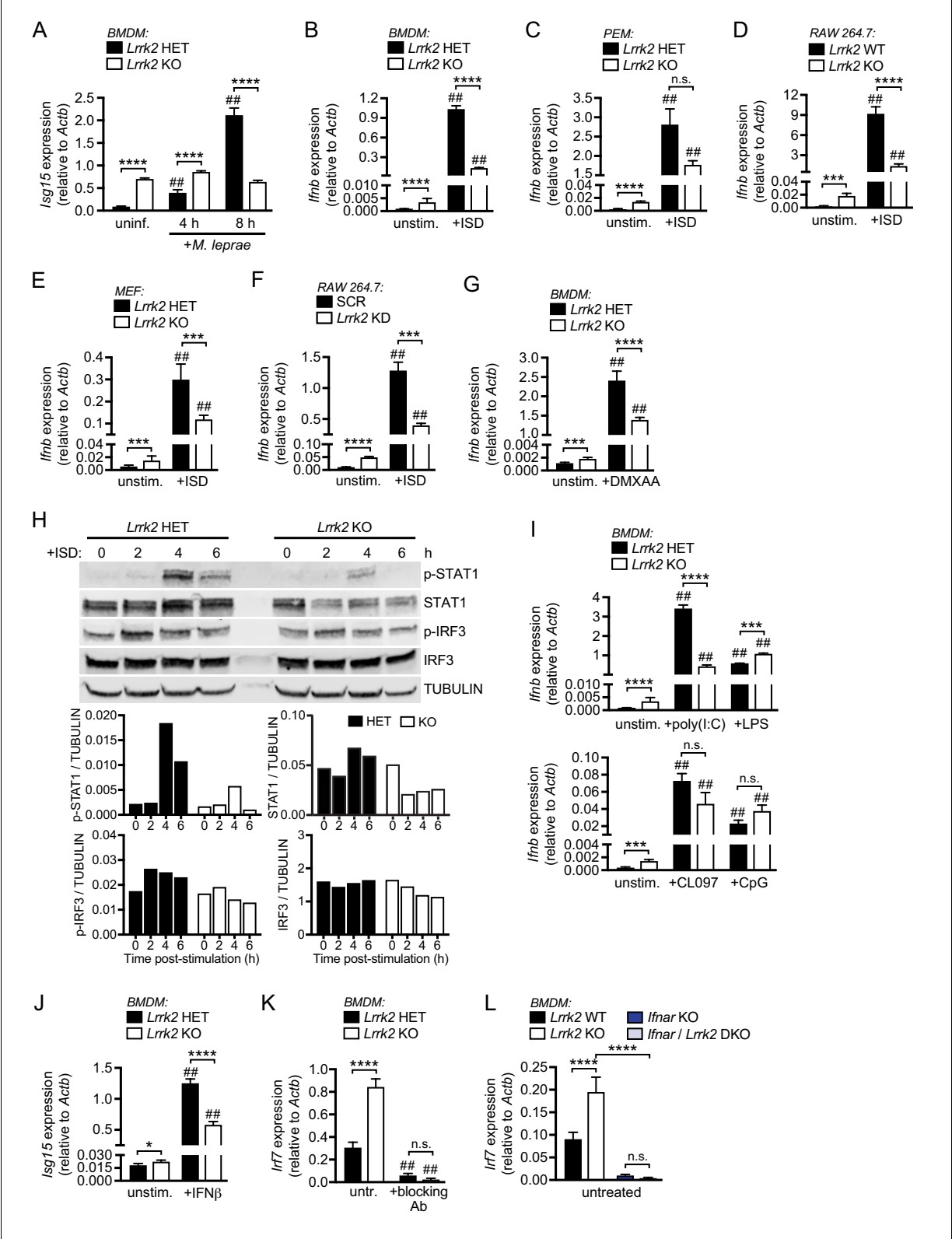

**Figure 2.** *Lrrk2* KO macrophages exhibit blunted type I IFN expression in response to cytosolic nucleic acid agonists. (**A**) RT-qPCR of *Isg15* expression after 4 and 8 hr of infection with *M. leprae* (MOI = 50) in *Lrrk2* KO BMDMs and HET controls. (**B**) RT-qPCR of *Ifnb* in unstimulated *Lrrk2* KO and HET BMDMs alongside cells transfected with 1 µg/ml ISD (dsDNA) for 4 hr. (**C**) As in (**B**) but with peritoneal macrophages (PEMs) from *Lrrk2* KO and HET mice elicited for 4 days with 1 ml 3% Brewer's thioglycolate broth. (**D**) As in (**B**) but with RAW 264.7 *Lrrk2* KO cells and WT controls. (**E**) As in (**B**) but with

*Figure 2 continued on next page*

*Figure 2 continued*

MEFs from day 14.5 *Lrrk2* KO or HET embryos. (**F**) As in (**B**) but with RAW 264.7 *Lrrk2* KD and scramble (SCR) controls cells. (**G**) RT-qPCR of *Ifnb* expression in uninfected *Lrrk2* KO or HET BMDMs and in cells treated with 50 ng/ml DMXAA for 2 hr. (**H**) Western blot analysis and quantification of IRF3 phosphorylation (Ser396) and STAT1 phosphorylation (Tyr701) in BMDMs from HET and *Lrrk2* KO mice compared to total IRF3 and STAT1 with tubulin as a loading control following transfection with 1 µg/ml ISD (dsDNA). (**I**) As in (**G**) but following transfection with 1 µg/ml poly(I:C), 100 ng/ml LPS, transfection with 10 µM CpG 2395, or stimulation with 1 µM CL097, all for 4 hr. (**J**) RT-qPCR of *Isg15* expression after treatment with 200 IU IFN-β for 4 hr. (**K**) RT-qPCR of *Irf7* gene expression in *Lrrk2* HET and KO BMDMs with or without overnight treatment with IFN-β neutralizing antibody (blocking Ab, 1:250). (**L**) RT-qPCR of *Irf7* gene expression in BMDMs from WT, *Lrrk2* KO, *Ifnar* KO, and double knockout (*Lrrk2/Ifnar* DKO) mice. Statistical analysis: *p<0.05, **p<0.01, ***p<0.005, ****p<0.001 (comparing indicated data points); ##p<0.001 (comparing stimulated to unstimulated of same genotype). (**A–L**) two-way ANOVA Tukey post-test.

The online version of this article includes the following figure supplement(s) for figure 2:

**Figure supplement 1.** LRRK2-deficient macrophages are unable to properly induce type I IFN expression in response to nucleic acid agonists.

1C), suggesting that TLR responses are intact in the absence of LRRK2 but cytosolic DNA and RNA sensing pathways are perturbed. We observed similar phenotypes for PEMs and MEFs treated with LPS (*Figure 2—figure supplement 1D–E*) and poly(I:C) (*Figure 2—figure supplement 1E*). *Lrrk2* KO BMDMs were, however, defective in ISG expression following recombinant bioactive IFN-β treatment (which directly engages with IFNAR) (*Figure 2J* and *Figure 2—figure supplement 1F*).

Because *Lrrk2* KO BMDMs failed to induce ISG expression following IFN-β treatment, we hypothesized that the elevated basal levels of type I IFN transcripts prevented *Lrrk2* KO macrophages from inducing a response at the level of IFNAR. To begin to test this prediction, we wanted to see if blocking IFN-β engagement with IFNAR could break this loop and 'reset' basal ISGs in *Lrrk2* KO macrophages. Indeed, when HET and KO *Lrrk2* BMDMs were treated with an IFN-β neutralizing antibody, there was a reduction in basal levels of *Isg15* and *Irf7* in *Lrrk2* KO cells (*Figure 2K* and *Figure 2—figure supplement 1G*). We next tested if loss of IFNAR signaling could similarly rescue the *Lrrk2* KO phenotype by crossing *Lrrk2* KO mice to *Ifnar* KO mice. In *Lrrk2/Ifnar* double KO BMDMs, we also observed a significant reduction in basal ISG levels (*Figure 2L* and *Figure 2—figure supplement 1H*). These results provide additional evidence that the type I IFN program is chronically engaged in *Lrrk2* KO macrophages.

## Increased basal type I IFN in *Lrrk2* KO macrophages is dependent on cytosolic DNA sensing through cGAS

Because both IFN-β blockade and loss of *Ifnar* normalized basal ISG expression in *Lrrk2* KO macrophages, we hypothesized that *Lrrk2* contributes to basal type I IFN expression upstream of cytosolic RNA (MAVS/RIG-I) or DNA (cGAS/STING) sensing, two nucleic acid sensing pathways that are interconnected between positive and negative feedback loops (*Zevini et al., 2017*). To directly test the involvement of cGAS in generating elevated resting levels of type I IFN in *Lrrk2* KO macrophages, we crossed *Lrrk2* KO and *cGas* KO mice and compared type I IFN transcript levels in double KO BMDMs with those of littermate controls. Although basal *Isg15* expression differences between *Lrrk2* KO and HET BMDMs were more modest in this experiment, loss of cGAS significantly reduced basal ISG expression in *Lrrk2* KO BMDMs (*Figure 3A* and *Figure 3—figure supplement 1A*). With lowered resting type I IFN levels, *cGas/Lrrk2* double KOs were able to respond normally to IFN/ISG-generating innate immune stimuli like DMXAA, which bypasses cGAS and stimulates STING directly (*Diner et al., 2013*), and poly(I:C) transfection (*Figure 3A* and *Figure 3—figure supplement 1A*). Consistent with the ability of *cGas* ablation to rescue *Lrrk2* KO baseline and induction defects, western blot analysis showed that levels of STAT1 phosphorylation were restored in *cGas/Lrrk2* double KOs (*Figure 3B*). Together, these results support a model where high basal levels of type I IFN and ISGs in *Lrrk2* KO macrophages are due to chronic engagement of the cGAS-dependent DNA sensing pathway.

## Cytosolic sensing of mtDNA contributes to basal type I IFN expression in *Lrrk2* KO macrophages

We next sought to identify the source of the chronic cGAS-activating signal. Mitochondrial DNA (mtDNA) has been shown to be a potent activator of type I IFNs downstream of cGAS (*West et al., 2015*), and LRRK2 is known to influence mitochondrial homeostasis (*Singh et al., 2019*), albeit

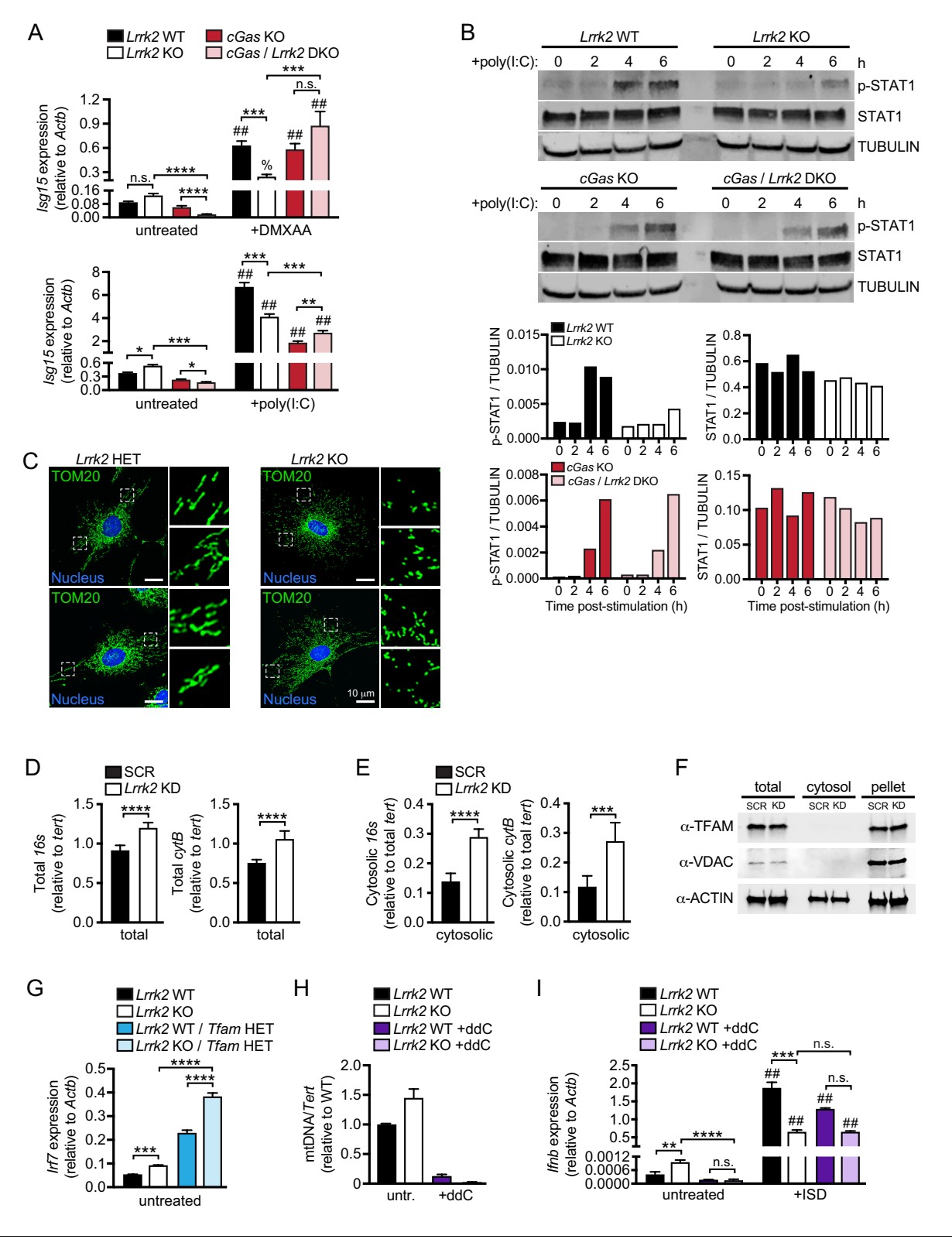

**Figure 3.** Cytosolic mtDNA drives basal type I IFN expression in *Lrrk2* KO macrophages. (**A**) *Isg15* gene expression in *Lrrk2* WT, *Lrrk2* KO, *cGas* KO, and double KO (*cGas/Lrrk2* DKO) BMDMs treated with 5 μg/ml DMXAA or transfected with 1 μg poly(I:C) for 4 hr. (**B**) Western blot analysis of STAT1 phosphorylation (Tyr701) in BMDMs from WT, *Lrrk2* KO, *cGas* KO, and *cGas/Lrrk2* double knockout (DKO) mice compared to total STAT1 with tubulin as a loading control. (**C**) Immunofluorescent images with anti-TOM20 antibody to visualize the mitochondrial network of *Lrrk2* HET and KO MEFs.
*Figure 3 continued on next page*

Figure 3 continued

TOM20 (green), nucleus (DAPI, blue); Scale bar = 10 μm (D) qPCR of total *16*s and *cytB* (mitochondrial DNA) relative to *Tert* (nuclear DNA). (E) As in (D) but cytosolic mitochondrial DNA. (F) Western blot of ACTIN, TFAM, and VDAC protein levels in total, cytosol, and pellet (organelle and membrane) fractions of *Lrrk2* KD and SCR RAW 264.7 cells. (G) *Irf7* gene expression normalized to *Actb* in untreated BMDMs from *Lrrk2* WT, *Lrrk2* KO, *Tfam* HET, and *Lrrk2* KO/*Tfam* HET mice. (H) qPCR of dLoop (mitochondrial DNA) normalized to *Tert* (nuclear) to confirm mtDNA depletion in WT and *Lrrk2* KO RAW 264.7 cells treated with 10 μM ddC for 4 days. (I) RT-qPCR of *Ifnb* gene expression in WT and *Lrrk2* KO RAW 264.7 cells with or without ddC treatment, untreated and at 4 hr post-transfection with 1 μg/ml ISD. Statistical analysis: *p<0.05, **p<0.01, ***p<0.005, ****p<0.001 (comparing indicated data points); %p<0.05, ##p<0.001 (comparing stimulated to unstimulated of same genotype). (A and I) 3-way ANOVA, Tukey post-test; (D and E) two-tailed Student's T test; (G and H) two-way ANOVA Tukey post-test.

The online version of this article includes the following figure supplement(s) for figure 3:

**Figure supplement 1.** Higher levels of cytosolic mtDNA contribute to defective type I IFN responses in Lrrk2 KO macrophages.

through mechanisms that are not entirely clear. To begin implicating mtDNA in the dysregulation of type I IFNs in *Lrrk2* KO cells, we first investigated the status of the mitochondrial network in *Lrrk2* HET and KO MEFs. As previously described for cells overexpressing wild-type or mutant alleles of *LRRK2* (*Yang et al., 2014*), *Lrrk2* KO MEFs had a more fragmented mitochondrial network, especially around the cell periphery, as evidenced by punctate TOM20 staining (*Figure 3C*). We hypothesized that this fragmentation was a sign of mitochondrial damage that could allow mitochondrial matrix components, including mtDNA, to leak into the cytosol. Therefore, we isolated the cytosolic fraction of control and *Lrrk2* KD RAW 264.7 macrophages (*Figure 3D–F*) and *Lrrk2* HET and KO MEFs (*Figure 3—figure supplement 1B–C*) and measured cytosolic mtDNA abundance. We found that although *Lrrk2* KD cells had only slightly higher total mtDNA compared to controls (*Figure 3D*), they had ~3 fold more cytosolic mtDNA (*Figure 3E*). A similar phenotype was seen with *Lrrk2* KO MEFs (*Figure 3—figure supplement 1B-C*). This increase in cytosolic mtDNA was not simply an artifact of fragmented mitochondria contaminating cytosolic fractions as neither TFAM, an abundant mitochondrial transcription factor, nor VDAC, a mitochondrial outer membrane protein, were detectable in the cytosolic fraction by western blot (*Figure 3F*).

We hypothesized that cytosolic mtDNA results in activation of cGAS/IFNAR signaling, which ultimately limits the ability of *Lrrk2* KO macrophages to respond to additional cytosolic nucleic acid agonists by downregulating canonical IFNAR signaling (consistent with a reduction in STAT1 phosphorylation (*Figure 2H*)). To exacerbate the proposed mitochondrial defect, we crossed *Lrrk2* KO mice with *Tfam* HET mice. *Tfam* HET mice are deficient in the mitochondrial transcription factor required for maintaining the mtDNA network and thus have high levels of cytosolic mtDNA (*Kasashima et al., 2011*; *West et al., 2015*). Depleting TFAM in *Lrrk2* KO BMDMs led to even higher basal ISG expression (*Figure 3G*), further suggesting that release of mtDNA into the cytosol in *Lrrk2* KO cells contributes to their elevated type I IFN expression. We next sought to rescue type I IFN defects in *Lrrk2* KO macrophages by depleting mtDNA using ddC (2′,3′-dideoxycytidine), an inhibitor of mtDNA synthesis (*Leibowitz, 1971*; *Meyer and Simpson, 1969*). Treating *Lrrk2* KO RAW 264.7 cells with ddC substantially reduced mtDNA copy number (*Figure 3H*) and resulted in similar basal expression of type I IFN and ISGs in resting *Lrrk2* HET and KO cells (*Figure 3I* and *Figure 3—figure supplement 1D*). Importantly, when mtDNA-depleted *Lrrk2* KO RAW 264.7 macrophages were stimulated with ISD, their ability to induce *Ifnb* was restored to that of wild-type; *Lrrk2* KO macrophages induced *Ifnb* approximately 500-fold in the absence of ddC but approximately 5000-fold following ddC treatment while WT macrophages induced *Ifnb* between 4000–5000-fold +/- ddC treatment (*Figure 3I* and *Figure 3—figure supplement 1D-E*). These results demonstrate a critical role for mtDNA in driving both the high basal levels of type I IFN and the inability to properly induce type I IFN expression in *Lrrk2* KO macrophages.

## Defects in type I IFN responses in *Lrrk2* KO macrophages are due, in part, to increased DRP1 phosphorylation and mitochondrial fragmentation

Previous studies of microglia have shown that LRRK2 contributes to mitochondrial homeostasis through interaction with the mitochondrial fission protein DRP1 (*Ho et al., 2018*). Thus, we hypothesized that the loss of LRRK2 may compromise mitochondrial stability via misregulation of DRP1 activity, leading to fragmented mitochondria and spillage of mtDNA into the cytosol. To assess gross

defects in DRP1 distribution in the absence of LRRK2, we performed immunofluorescence microscopy and did not observe any obvious, qualitative changes to the expected distribution of DRP1 at the ends of mitochondrial tubules in *Lrrk2* KO MEFs, although TOM20 staining again revealed extensive fragmentation of the peripheral network (*Figure 4—figure supplement 1A*). We next asked whether DRP1 activity was impacted by loss of LRRK2. Because DRP1 is known to be positively regulated via phosphorylation at Ser616 (*Taguchi et al., 2007*), we performed flow cytometry with an antibody specific for phospho-S616 DRP1 and observed significantly higher levels of phospho-S616 DRP1 in *Lrrk2* KD RAW 264.7 cells, *Lrrk2* KO BMDMs, and *Lrrk2* KO MEFs compared to controls (*Figure 4A,C,D*). Western blot analysis of phospho-S616 DRP1 confirmed a modest increase in *Lrrk2* KD cells, while total DRP1 protein levels remained unchanged (*Figure 4B* and *Figure 4—figure supplement 1B*). Accumulation of phospho-S616 DRP1 was enhanced in MEFs by the addition of $H_2O_2$, which induces DRP1-dependent mitochondrial fission, and was eliminated with the addition of Mdivi-1, a specific inhibitor of DRP1 (*Figure 4—figure supplement 1C*).

Next, to test if DRP1 activity was linked to high basal type I IFN/ISG expression in *Lrrk2* deficient cells, we chemically inhibited DRP1 with Mdivi-1 and measured basal gene expression levels. In *Lrrk2* KD RAW 264.7 macrophages and *Lrrk2* KO BMDMs, DRP1 inhibition returned ISG expression to control levels (*Figure 4E* and *Figure 4—figure supplement 1D*). DRP1 inhibition also restored the cytosolic mtDNA levels in *Lrrk2* KD RAW 264.7 cells and *Lrrk2* KO MEFs to those of control cells (*Figure 4F* and *Figure 4—figure supplement 1E*). Together, these data indicate that dysregulated ISG expression in *Lrrk2* KO macrophages is caused by leakage of mtDNA into the cytosol, which occurs downstream of excessive DRP1-dependent mitochondrial fission.

## *Lrrk2* KO macrophages are susceptible to mitochondrial stress and have altered cellular metabolism

Given that cytosolic mtDNA contributes to type I IFN defects in *Lrrk2* KO macrophages, we predicted that mitochondria in *Lrrk2* KO cells may be more damaged and/or more prone to damage. To better understand the health of the mitochondrial network in *Lrrk2* KO vs. HET BMDMs, we first used the carbocyanine dye JC-1, which accumulates in mitochondria to form red fluorescent aggregates. Upon loss of mitochondrial membrane potential, JC-1 diffuses into the cytosol where it emits green fluorescence as a monomer. Thus, a decrease in red fluorescence (aggregates) and increase in green fluorescence (monomers) signifies mitochondrial depolarization, making JC-1 dye a highly sensitive probe for mitochondrial membrane potential. Flow cytometry analysis of resting *Lrrk2* HET and KO cells revealed lower levels of JC-1 dye aggregation (i.e., lower mitochondrial membrane potential) in *Lrrk2* KO BMDMs (*Figure 5A–B*), *Lrrk2* KD RAW 264.7 macrophages (*Figure 5—figure supplement 1A*), and primary *Lrrk2* KO MEFs (*Figure 5—figure supplement 1B*), compared to control cells. A baseline reduction in membrane potential was also detected using TMRE (tetramethylrhodamine, ethyl ester, perchlorate), a cell-permeable dye that is readily sequestered by active (positively charged) mitochondria, in *Lrrk2* KO BMDMs, *Lrrk2* KO MEFs, and *Lrrk2* KD RAW 264.7 cells (*Figure 5—figure supplement 1D*). In addition, *Lrrk2* KO BMDMs were more sensitive to the mitochondrial damaging and depolarizing agents, rotenone/ATP and FCCP, as measured by both JC-1 (*Figure 5C*; RAW 264.7 *Lrrk2* KDs and *Lrrk2* KO MEFs in *Figure 5—figure supplement 1A and B*, respectively) and TMRE (*Figure 5D* and *Figure 5—figure supplement 1C, E*). Interestingly, the mitochondrial membrane potential of *Lrrk2* KO BMDMs was normalized after treatment with Mdivi-1 to inhibit DRP1 (*Figure 5E–F*), suggesting that misregulation of DRP1 occurs upstream of LRRK2-dependent defects in mitochondrial membrane potential.

Previous reports have indicated that LRRK2 dysfunction alters ROS levels (*Pereira et al., 2014*; *Russo et al., 2019*). To test whether ROS could contribute to the defective type I IFN signature in *Lrrk2* KO cells, we treated *Lrrk2* HET and KO BMDMs with mitoTEMPO (mitoT), a mitochondrially-targeted scavenger of superoxide (*Liang et al., 2010*). Consistent with oxidative stress driving misregulation of the type I IFN response in the absence of LRRK2, we observed a dramatic rescue of basal ISG expression in *Lrrk2* KO cells treated with mitoT (*Figure 5G*). Together these data provide strong evidence that *Lrrk2* KO cells harbor a baseline defect in mitochondrial membrane potential, likely due to DRP1 activation, that results in chronic type I IFN induction due to increased levels of cytosolic mtDNA.

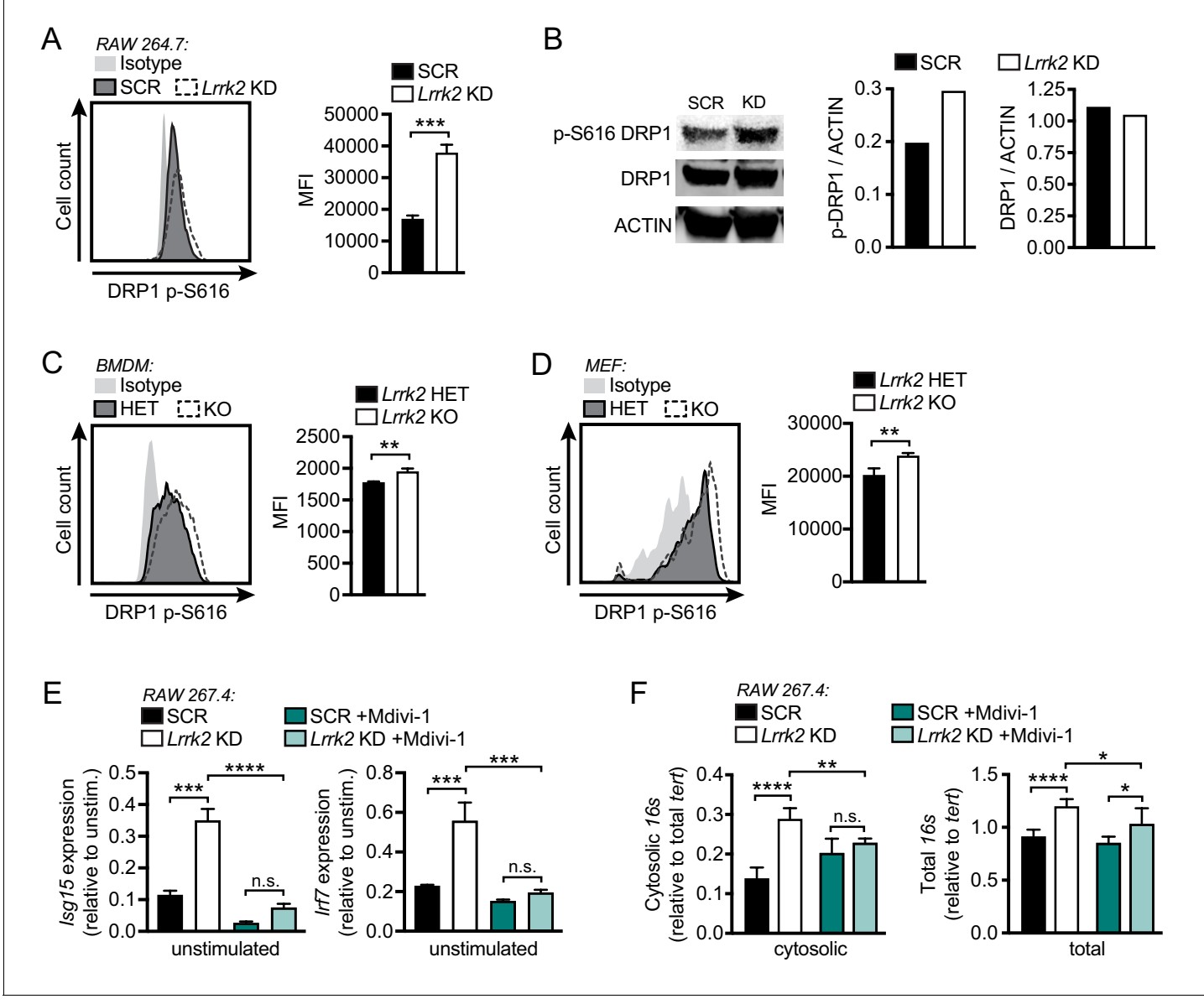

**Figure 4.** Mitochondrial fragmentation and increased DRP1 phosphorylation contribute to type I IFN dysregulation in *Lrrk2* KO macrophages. (**A**) Histograms showing counts of phospho-S616-DRP1 in SCR and *Lrrk2* KD RAW 264.7 cells as measured by flow cytometry. (**B**) Western blot analysis and quantification of DRP1 phosphorylation (Ser616) in SCR and *Lrrk2* KD RAW 264.7 cells compared to total DRP1 and actin as a loading control. (**C**) As in (**A**) but for BMDMs. (**D**) As in (**A**) but for MEFs. (**E**) Basal gene expression of *Isg15* and *Irf7* in SCR and *Lrrk2* KD RAW 267.4 cells treated with Mdivi-1 50 µM for 12 hr. (**F**) qPCR of cytosolic and total *16s* (mitochondrial DNA) relative to *Tert* (nuclear DNA) in SCR and *Lrrk2* KD RAW 264.7 cells treated with 50 µM Mdivi-1 for 12 hr. Statistical analysis: *p<0.05, **p<0.01, ***p<0.005, ****p<0.001. (**A, C,** and **D**) Two-tailed Student's T test; (**E** and **F**) two-way ANOVA Tukey post-test.

The online version of this article includes the following figure supplement(s) for figure 4:

**Figure supplement 1.** Hyperphosphorylation of DRP1 contributes to defects in type I IFN induction inLrrk2 KO macrophages.

## *Lrrk2* KO macrophages are defective in oxidative phosphorylation and glycolysis

Metabolic reprogramming is becoming increasingly appreciated as a critical contributor to macrophage polarization and transcriptional output (*Angajala et al., 2018*; *Sancho et al., 2017*). We hypothesized that mitochondrial defects may render *Lrrk2* KO macrophages incapable of meeting metabolic demands. To test this idea, we manipulated levels of sodium pyruvate, an intermediate

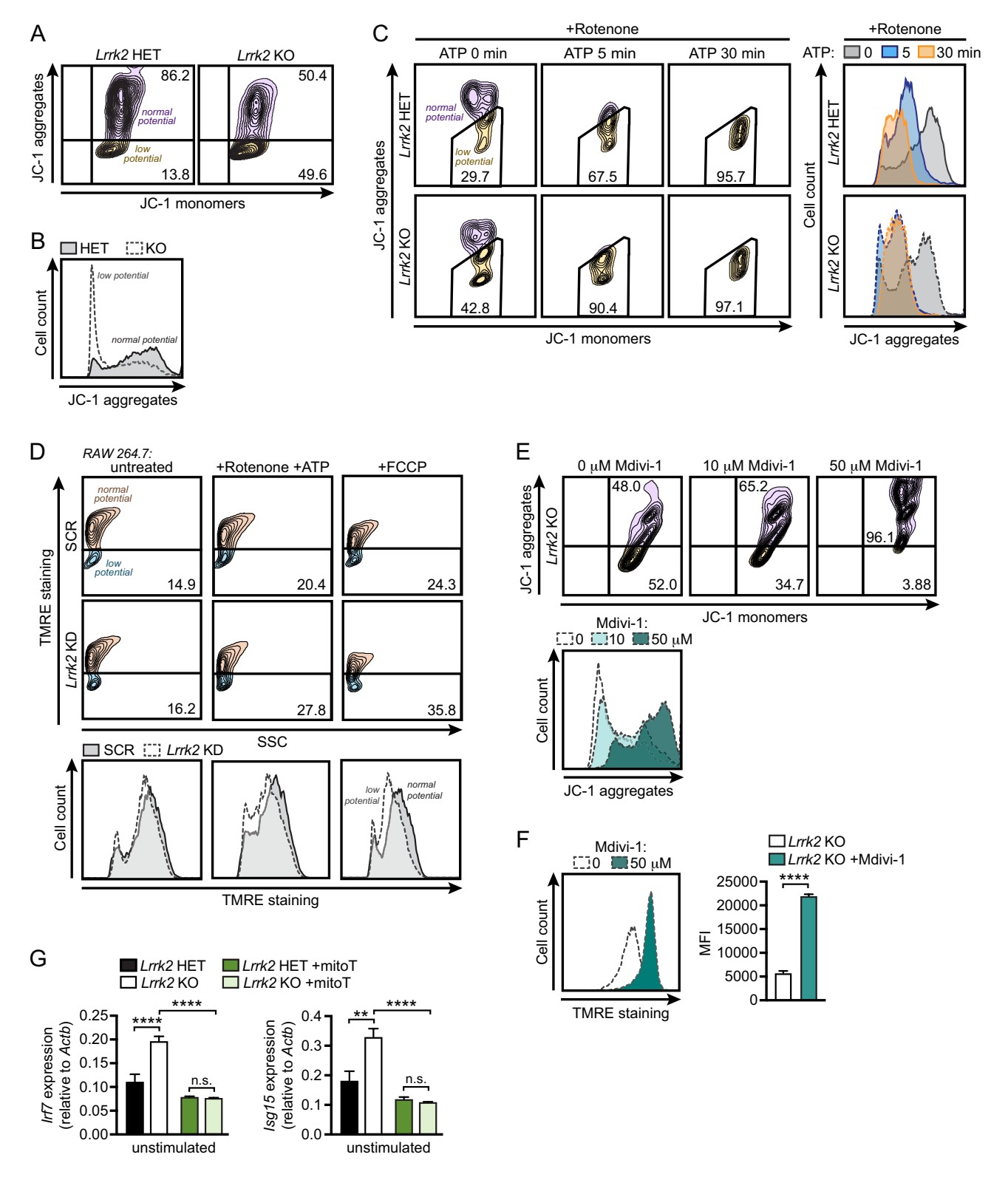

**Figure 5.** *Lrrk2* KO macrophages are more susceptible to mitochondrial stress. (**A**) Mitochondrial membrane potential in *Lrrk2* HET and KO BMDMs as measured by JC-1 dye by flow cytometry. Aggregates (610/20) indicate normal mitochondrial membrane potential and monomers (520/50) indicate low membrane potential. (**B**) Histogram of (**A**) displaying cell counts of JC-1 aggregates for *Lrrk2* HET and KO BMDMs. (**C**) JC-1 aggregates measured by flow cytometry in BMDMs treated for 3 hr with 2.5 μM rotenone followed by 5 μM ATP for 0, 5, or 30 min. Histogram of cell counts is on the right. (**D**)

*Figure 5 continued on next page*

*Figure 5 continued*

Flow cytometry of mitochondrial membrane potential measured by TMRE (585/15) in SCR and *Lrrk2* KD RAW 264.7 cells treated for 3 hr with 2.5 μM rotenone followed by 5 μM ATP for 15 min or 50 μM FCCP for 15 min. (E) JC-1 aggregates measured by flow cytometry in *Lrrk2* KO BMDMs treated with 10 or 50 μM Mdivi-1 for 4 hr. (F) The same as in (E) but with TMRE. (G) Basal gene expression of *Irf7* and *Isg15* in *Lrrk2* HET and KO BMDMs treated overnight with 200 μM mitoTEMPO. JC-1 flow cytometry assays are representative of 3 independent experiments. Statistical analysis: **p<0.01, ***p<0.005, ****p<0.001. (F) Two-tailed Student's T test; (G) two-way ANOVA Tukey post-test.

The online version of this article includes the following figure supplement(s) for figure 5:

**Figure supplement 1.** Lrrk2 KO macrophages are more prone in mitochondrial depolarization in response to mitochondrial stresses.

metabolite of glycolysis and the TCA cycle, in the media of *Lrrk2*-de*ficient* cells plus their respective controls. The presence of pyruvate places additional demands on the mitochondria by pushing cells towards oxidative metabolism rather than glycolysis. Remarkably, addition of as little as 1 mM sodium pyruvate to the growth media increased the already high basal levels of type I IFN in macrophages and MEFs lacking LRRK2 (*Figure 6A–B* and *Figure 6—figure supplement 1A*), suggesting that increasing metabolic demands on mitochondria promotes further leakage of mtDNA into the cytosol in these cells.

To better understand how *Lrrk2* KO macrophages may be defective in meeting the energy needs of the cell, we used the Agilent Seahorse Metabolic Analyzer to measure cellular respiration. In this assay, oxidative phosphorylation (OXPHOS) and glycolysis are assayed by oxygen consumption rate (OCR) and extracellular acidification rate (ECAR), respectively. We found that mitochondria in *Lrrk2* KO BMDMs were defective in both maximal and reserve respiratory capacity (*Figure 6C*, top panels), indicating reduced OXPHOS. *Lrrk2* KO macrophages were also defective in non-glycolytic acidification and had reduced glycolysis. (*Figure 6C*, bottom panels). This result was surprising as macrophages typically switch from OXPHOS to glycolysis when activated (*Kelly and O'Neill, 2015*), but *Lrrk2* KO macrophages have reduced utilization of both energy producing pathways. Remarkably, co-treatment of *Lrrk2* KO BMDMs with mitoT and IFN-β neutralizing antibody completely restored OXPHOS and glycolysis. This rescue was greater than treatment of either IFN-β blockade or mitoT alone (*Figure 6C–D*), indicating that mitochondrial ROS and constitutive IFNAR signaling independently contribute to the metabolic defects in *Lrrk2* KO macrophages. Conversely, when *Lrrk2* KO BMDMs were cultured in the presence of increasing concentrations of sodium pyruvate (1 and 2 mM), we observed exacerbated metabolic defects in OXPHOS and glycolysis (*Figure 6—figure supplement 1B-C*). Collectively, these data demonstrate that loss of LRRK2 in macrophages has a profound impact on the mitochondria, not only promoting their fragmentation, but also rendering them less capable of utilizing different carbon sources and meeting the energy needs of the cell.

## Reduced antioxidants and purine metabolites contribute to mitochondrial damage and type I IFN expression in *Lrrk2* KO macrophages

To better understand possible molecular changes driving or resulting from damaged mitochondria in *Lrrk2* KO macrophages, we performed an unbiased query of metabolites using LC/MS/MS (Table S2). In *Lrrk2* KO BMDMs, we found lower levels of inosine monophosphate (IMP) and hypoxanthine, two intermediates in the purine biosynthesis pathway, which we validated using pure molecular weight standards (*Figure 7A* and *Figure 7—figure supplement 1A-B*). Interestingly, purine metabolism is tightly associated with generation of antioxidant compounds, and several metabolites in this pathway are well-characterized biomarkers of Parkinson's disease (*Figure 7B*; *Zhou et al., 2012*). Consistent with lower levels of antioxidants, we detected increased oxidized glutathione and glutamate metabolism compounds in *Lrrk2* KO macrophages (Table S2). Moreover, in accordance with lower levels of purine metabolites, we observed significantly fewer de novo biosynthesis puncta containing formylglycinamidine ribonucleotide synthase (FGAMS, also known as PFAS), a core purinosome component, per *Lrrk2* KO MEF cell compared to HET controls (*Figure 7C–D*).

Because depleted antioxidant pools and concomitant accumulation of ROS can lead to mitochondrial damage, we hypothesized that ROS might contribute to the mitochondrial and type I IFN defects we observed in *Lrrk2* KO macrophages. To test this, we first supplemented cells with antioxidants in order to rescue the type I IFN defect in *Lrrk2* KO macrophages. Addition of urate, a

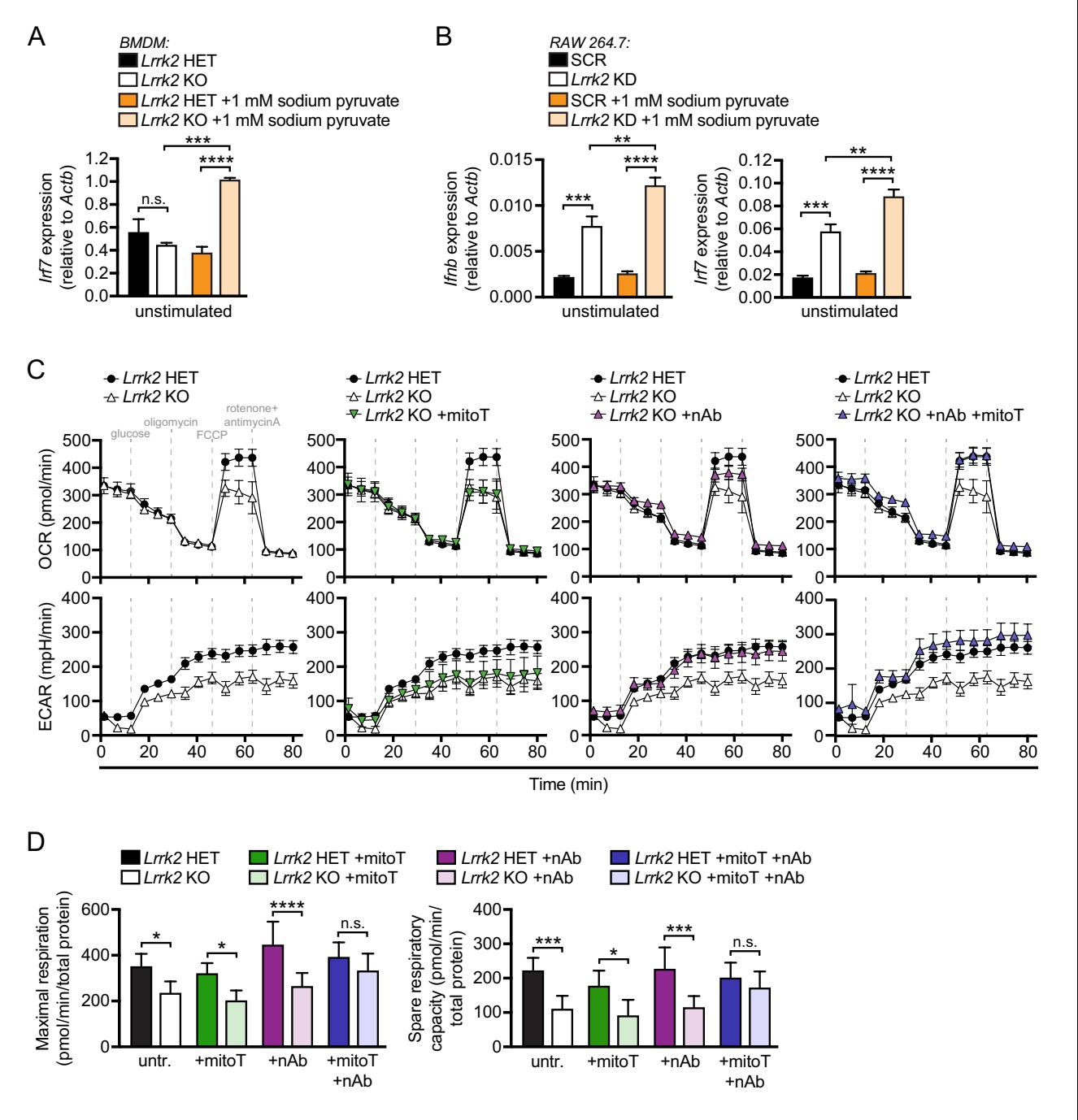

**Figure 6.** *Lrrk2* KO macrophages are defective in oxidative phosphorylation and glycolysis. (**A**) *Irf7* gene expression in HET and *Lrrk2* KO BMDMs cultured with or without 1 mM sodium pyruvate. (**B**) *Ifnb* and *Irf7* gene expression in SCR and *Lrrk2* KD RAW 264.7 cells cultured with or without 1 mM sodium pyruvate. (**C**) BMDMs from *Lrrk2* HET and KO mice were treated with 200 µM mitoTEMPO, IFN-β blocking antibody, and the combination of both overnight followed by analysis of oxygen consumption rate (OCR) and extracellular acidification rate (ECAR) measured using the Seahorse Metabolic Analyzer (Agilent). (**D**) Quantification of maximal respiration and spare respiratory capacity from (**C**). Statistical analysis: *p<0.05, **p<0.01, ***p<0.005, ****p<0.001. (A, B, and D) two-way ANOVA Tukey post-test.

The online version of this article includes the following figure supplement(s) for figure 6:

**Figure supplement 1.** Increasing sodium pyruvate concentrations increases basal Ifnb expression and exacerbates metabolic defects in Lrrk2 KO macrophages.

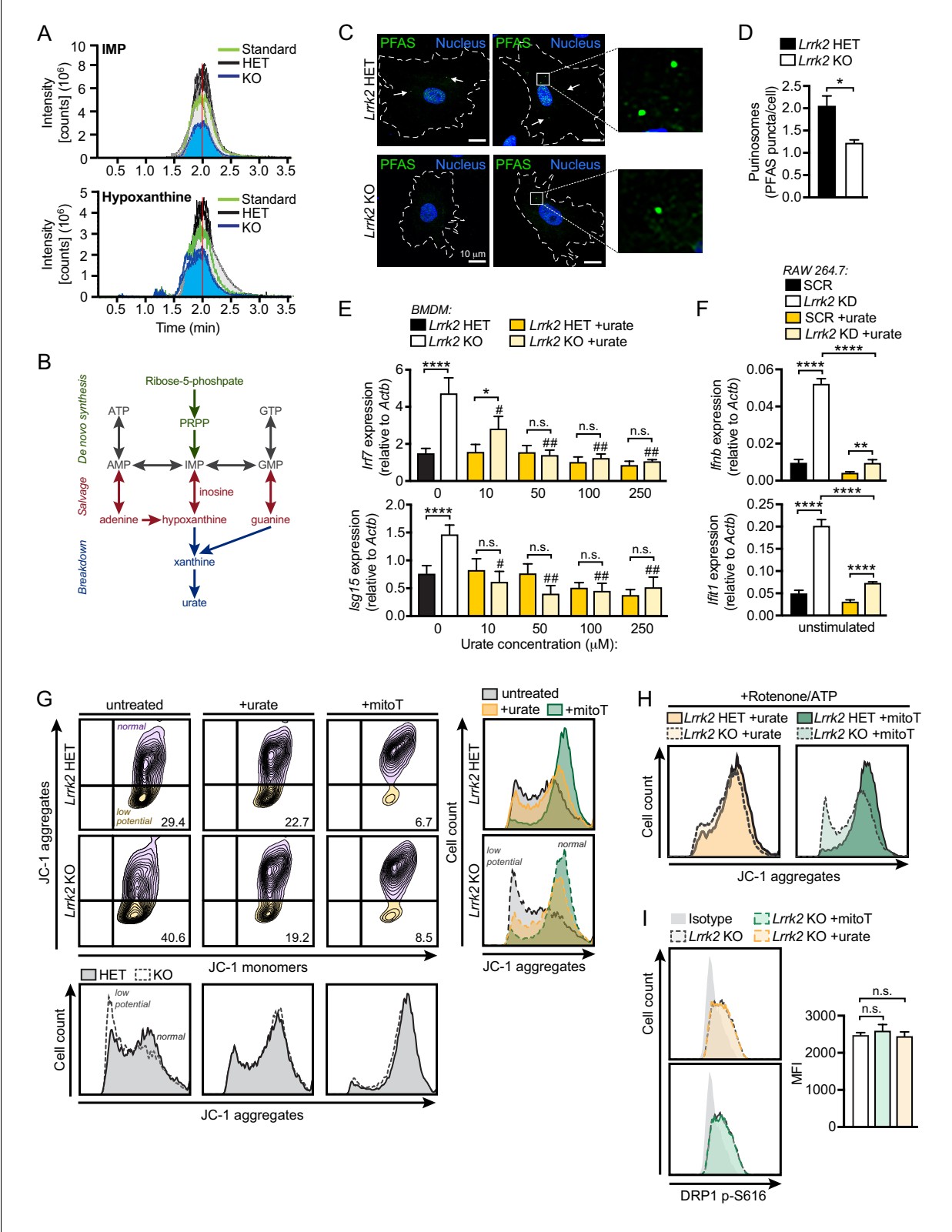

**Figure 7.** Reduced antioxidant pools in *Lrrk2* KO macrophages result in mitochondrial stress. (**A**) Chromatogram depicting targeted metabolomic analysis of *Lrrk2* HET (n = 3) and KO BMDMs (n = 3) with pure molecular weight standard to IMP (top) and hypoxanthine (bottom). Replicate experiments are shown as individual lines (n = 2). Coefficient of variance (CV) for IMP = 8.8% (KO) and 21.7% (HET). CV for hypoxanthine = 9.1% (KO) and 14.0% (HET). (**B**) Diagram of key metabolites produced during purine metabolism oriented to the major steps of the pathway. De novo synthesis

*Figure 7 continued on next page*

*Figure 7 continued*

(green), salvage (red), breakdown (blue). (**C**) Representative immunofluorescence microscopy image of purinosome formation measured by PFAS puncta (green) in *Lrrk2* HET and KO MEFs. Nuclei stained with DAPI (blue). (**D**) Quantification of number of PFAS puncta per cell. 100 cells were counted per coverslip from three coverslips. (**E**) RT-qPCR of *Irf7* and *Isg15* gene expression in *Lrrk2* HET and KO BMDMs treated with increasing concentrations of urate (10, 50, 100, and 250 µM for 24 hr). (**F**) Basal gene expression of *Ifnb* and *Ifit1* in SCR and *Lrrk2* KD RAW 264.7 cells treated with 250 µM urate overnight. (**G**) JC-1 aggregate vs. monomer formation measured by flow cytometry in *Lrrk2* HET and KO BMDMs treated with 100 µM urate or 200 µM mitoTEMPO overnight. Histograms shown below and merged histograms shown to the right. (**H**) Histograms of *Lrrk2* HET and KO BMDMs treated with 100 µM urate or 200 µM mitoTEMPO overnight and then treated with 2.5 µM rotenone for 3 hr followed by 5 µM ATP for 15 min. (**I**) Histograms of DRP1 p-S616 flow cytometry analysis for *Lrrk2* KO BMDMs following treatment with 100 µM urate or 200 µM mitoTEMPO. Quantification is shown on the right. JC-1 flow cytometry assays are representative of three independent experiments. Statistical analysis: \*p<0.05, \*\*p<0.01, \*\*\*p<0.005, \*\*\*\*p<0.001 (comparing indicated data points); #p<0.005, ##p<0.001 (comparing treated to untreated of same genotype). (**D**) Two-tailed Student's T test; (**E and F**) two-way ANOVA Tukey post-test; (**I**) one-way ANOVA Tukey post-test.

The online version of this article includes the following figure supplement(s) for figure 7:

**Figure supplement 1.** LC-MS/MS analysis identifies lower levels of IMP and hypoxanthine in Lrrk2 KO macrophages.

free radical scavenger and major breakdown product of purine metabolism (*Figure 7B*), reduced basal ISG expression in a dose-dependent fashion in *Lrrk2* KO BMDMs and in *Lrrk2* KD RAW 264.7 cells (*Figure 7E–F*, respectively). Furthermore, treatment with urate or mitoT restored the resting mitochondrial membrane potential of *Lrrk2* KO BMDMs (*Figure 7G–H*), suggesting that radical oxygen species contribute to mitochondrial depolarization in the absence of LRRK2. Neither urate nor mitoT was sufficient to alter DRP1 activation in *Lrrk2* KO BMDMs, suggesting the antioxidant defects are either downstream or independent of LRRK-dependent DRP1 misregulation (*Figure 7I*). Collectively, these results suggest that the depletion of antioxidant pools in *Lrrk2* KO macrophages from defective purine metabolism contributes to their mitochondrial dysfunction and aberrant type I IFN expression.

## *Lrrk2* KO mice can control Mtb replication but have exacerbated infection-induced local inflammation

Previous reports have linked SNPs in *LRRK2* with susceptibility to mycobacterial infection and excessive inflammation in humans (*Fava et al., 2016*; *Wang et al., 2015*; *Zhang et al., 2009*). Our studies demonstrate that LRRK2 plays a key role in homeostasis of macrophages, the first line of defense and replicative niche of Mtb. Therefore, we sought to understand how LRRK2 deficiency influences Mtb pathogenesis in macrophages ex vivo and during an in vivo infection. *Lrrk2* HET and KO BMDMs were infected with Mtb (Erdman strain; MOI = 1), and colony forming units (CFUs) were measured over the course of five days. We observed a significant increase in CFUs recovered at 5 days (120 hr) following infection (*Figure 8A*), suggesting that while *Lrrk2* KO macrophages can control Mtb replication early after infection, they are more permissive once the bacteria have established a niche. These results are consistent with a recent report demonstrating that defective IFNAR signaling in BMDMs leads to increased Mtb replication (*Banks et al., 2019*).

To test whether this replication phenotype impacted Mtb pathogenesis in vivo, we infected *Lrrk2* HET and KO mice with ~150 CFUs via aerosol chamber delivery. We observed no difference in the survival time of Mtb-infected *Lrrk2* KO mice compared to HET controls (*Figure 8B*), and at Days 7, 21, 63, and 126 days post-infection, we observed no significant differences in bacterial burdens in the lungs or spleens of infected mice (*Figure 8C*). We also measured serum cytokines and found no major differences (*Figure 8D*). We next examined if local inflammation in the lungs was impacted by loss of LRRK2. While major NF-κB pathway inflammatory cytokines (e.g. *Tnfa*, *Il6*, *Il1b*) were expressed at similar levels in the lungs of Mtb-infected *Lrrk2* KO and HET mice at Day 21 (*Figure 8E*), we observed lower levels of several canonical ISGs including *Mx1*, *Irf9*, and *Gbp8* (*Figure 8F*), consistent with trends we observed in *Lrrk2* KO macrophages ex vivo (*Figure 1E*). These results began to suggest that during Mtb infection, most of the effects of LRRK2 ablation occur at the local site of infection.

To better understand the nature of this local inflammatory phenotype, we inspected lung tissues via hematoxylin and eosin (H&E) staining. We observed significantly more inflammatory granulomatous nodules in the lungs, specifically in the perivascular region (*Figure 8G–H*), indicating that more macrophages had infiltrated infected lungs of *Lrrk2* KO mice at Day 21 post-infection. Consistent

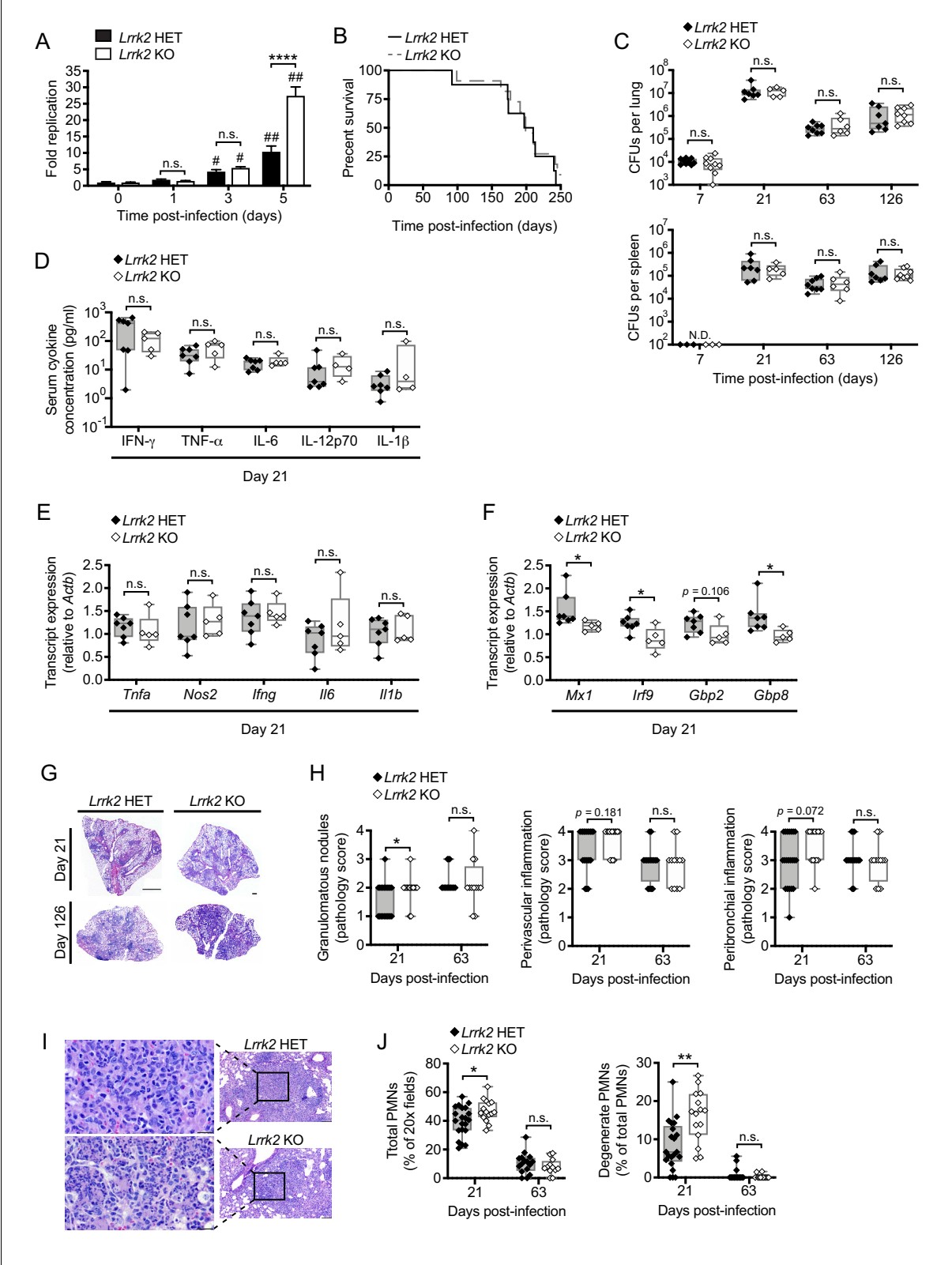

**Figure 8.** *Lrrk2* KO mice exhibit increased lung inflammation during Mtb infection. (**A**) Mtb colony forming units (CFUs) recovered from *Lrrk2* HET and KO BMDMs over the course of 5 days (MOI = 1). (**B**) Survival curves for *Lrrk2* HET (n = 8) and KO (n = 11) mice over a 250 day Mtb infection. Survival times not statistically different based on log-rank Mantel-Cox test. (**C**) CFUs recovered from lungs and spleens of Mtb-infected *Lrrk2* HET and KO mice at Day 7, 21, 63, and 126 post-infection. (**D**) Circulating serum cytokines measured at Day 21 in *Lrrk2* HET and KO mice. (**E**) RT-qPCR of inflammatory

*Figure 8 continued on next page*

*Figure 8 continued*

cytokines from total RNA recovered from lung homogenates from Day 21 Mtb-infected *Lrrk2* HET and KO mice. (**F**) As in (**E**) but detecting ISGs. (**G**) Hematoxylin and eosin (H&E) stain of inflammatory nodules in the lungs of *Lrrk2* KO and HET mice 21 days after infection with Mtb. Small scale bar, 500 µm; large scale bar 1 mm. (**H**) Semi-quantitative score of pulmonary inflammation with a score of 0, 1, 2, 3 or 4 assigned based on granulomatous nodules in none, up to 25%, 26–50%, 51–75% or 76–100% of fields, respectively. Perivascular and peribronchial inflammation was scored using an analogous scale based on percentage of medium-caliber vessels or bronchioles with adjacent inflammatory nodules. (**I**) H&E stain of neutrophils within an inflammatory nodule in the lung of *Lrrk2* HET and KO mice 21 days after infection with Mtb. Left panel bar is 20 µm. Right panel bar is 200 µm. (**J**) Quantification of neutrophils in the lungs of *Lrrk2* HET and KO mice infected with Mtb for 21 or 63 days. Total neutrophil scores were determined by the percentage of fields of view at 20X magnification containing neutrophils. Degenerate neutrophil scores were determined by the percentage of PMN positive fields containing degenerate neutrophils. Statistical analysis: *p<0.05, **p<0.01, ***p<0.005, ****p<0.001 (comparing indicated data points); #p<0.005, ##p<0.001 (comparing infected to uninfected of same genotype). (**A**) Two-way ANOVA Tukey post-test; (**B**) Mantel-Cox log-rank; (**C–J**) Mann-Whitney test.

with increased inflammation, we observed more total neutrophils (polymorphonuclear leukocytes, PMNs) as well as more PMNs undergoing cell death (degenerate PMNs) in the lungs of Mtb-infected *Lrrk2* KO mice compared to controls (*Figure 8I–J*). These results are consistent with a recently published study that reported enhanced inflammatory innate immune responses to Mtb infection in *Lrrk2* KO mice compared to wild-type controls (*Härtlova et al., 2018*).

## Discussion

Despite being repeatedly associated with susceptibility to mycobacterial infection and inflammatory disorders in genome-wide association studies, very little is known about how LRRK2 functions outside of the central nervous system. Here, we provide evidence that loss of LRRK2 in macrophages alters type I IFN and ISG expression due to elevated levels of cytosolic mtDNA and chronic cGAS signaling. During Mtb infection, loss of LRRK2 dysregulates type I IFN production and enhances local neutrophil and macrophage infiltration and cell death in the lung. These data help explain why *LRRK2* missense mutations are associated with exacerbated inflammation and poor disease outcomes in leprosy patients (*Fava et al., 2016*). They also hint at a previously unappreciated but potentially crucial role for LRRK2 in regulating the central nervous system immune milieu in PD patients (*Patrick et al., 2019*) via alteration of mitochondrial homeostasis in brain-resident glial cells. Our observations connect LRRK2's role in innate immune dysregulation with its requirement for maintaining mitochondrial homeostasis and are consistent with numerous recent studies linking mitochondrial metabolism and energy production to immune outcomes (*Angajala et al., 2018*; *Bird, 2019*; *Walker et al., 2014*).

There are several unique aspects of the *Lrrk2* KO macrophage phenotype that reveal new insights into how mitochondrial stress impacts type I IFN expression and innate immune cell priming. *Lrrk2* KO macrophages fail to activate normal levels of phospho-STAT1 following innate immune stimuli, resulting in blunted ISG induction, but this defect can be rescued by depleting mtDNA, reducing mitochondrial ROS, and deleting *cGas*. In this way, *Lrrk2* KO macrophages are phenotypically distinct from other cells in which mitochondrial stress and increased cytosolic mtDNA have been shown to dramatically increase phospho-STAT1, further amplifying the IFN response (*West et al., 2015*). We propose that cytosolic DNA and increased ROS— which perhaps together generate oxidized cytosolic DNA that is resistant to the exonuclease TREX-1 (*Gehrke et al., 2013*)—drive this unique refractory phenotype in *Lrrk2* KO macrophages. Alternatively or additionally, this phenomenon could be driven via by upregulation of one or more unidentified negative regulators of these key signaling pathways. Interestingly, *Lrrk2* KO macrophages also challenge the general paradigms of type I IFN priming. Typically, tonic or basal IFN levels are thought to 'rev the engine' so that cells can rapidly induce type I IFN expression after receiving a stimulus (*Taniguchi and Takaoka, 2001*). Using this metaphor, although the engine is revved in *Lrrk2* KO macrophages, these cells still lose the race. Additional studies will be needed to define the molecular mechanisms driving this puzzling phenotype.

The metabolic phenotypes of *Lrrk2* KO macrophages are also unique. It is curious that they suffer from defects in both glycolysis and oxidative phosphorylation, since generally, these two energy-producing pathways compensate for each other. Defects in both pathways is indicative of a more

quiescent cellular metabolic state consistent with a reduced capacity for IFNAR signaling. Importantly, treatment of *Lrrk2* KO macrophages with IFN-β neutralizing antibody was sufficient to rescue glycolysis (as measured by extracellular acidification rate (ECAR)) but not OXPHOS (as measured by oxygen consumption rate (OCR)) (*Figure 6C*), which suggests chronic IFNAR signaling alters the glycolytic rate in *Lrrk2* KO macrophages. Rescue of the OXPHOS defect required treatment with both IFN-β neutralizing antibody and mitoTEMPO, indicating that a more complex defect drives changes to mitochondrial respiration, perhaps linked to the depolarization defect. It will be important moving forward to understand the precise molecular contributions of LRRK2 to specific aspects of mitochondrial health (e.g. energy production, morphology, fission/fusion, etc.) and to link these defects with outcomes in diverse cell types (e.g. neurons and macrophages).

We propose that dysregulation of type I IFN expression in *Lrrk2* KO macrophages is the result of two distinct cellular defects conferred by loss of LRRK2. First, in the absence of LRRK2, decreased levels of purine metabolites and urate contribute to oxidative stress, leading to damage of the mitochondrial network. This idea is supported by our experiments showing that urate and mitoTEMPO treatments could rescue defects in mitochondrial polarization and return elevated basal type I IFNs to normal in *Lrrk2* KO macrophages (*Figures 5E* and *7E–G*). A recent human kinome screen identified LRRK2 as a kinase involved in dynamics of the purinosome, a cellular body composed of purine biosynthetic enzymes that assembles at or on the mitochondrial network (*French et al., 2016*). Specifically, shRNA knockdown of *LRRK2* in HeLa cells inhibited purinosome assembly and disassembly. As purinosomes are posited to form in order to protect unstable intermediates and increase metabolic flux through the de novo purine biosynthetic pathway (*An et al., 2008*; *Schendel et al., 1988*), we propose that the lower levels of IMP and hypoxanthine we measure in *Lrrk2* KO macrophages results from LRRK2-dependent defects in purinosome assembly (*Figure 7A*). Lower levels of purine nucleotide intermediates are especially notable in the context of PD; the plasma of PD patients (both *LRRK2* mutant-associated and idiopathic) has been shown to contain significantly less hypoxanthine and uric acid (*Johansen et al., 2009*), and patients with higher plasma urate levels, despite carrying *LRRK2* mutations, are less likely to develop PD (*Bakshi et al., 2019*). Furthermore, urate is currently being investigated as a potential therapeutic of PD, highlighting the importance of purine biosynthesis in this disease.

Second, we propose that loss of LRRK2 contributes to type I IFN dysregulation through defects associated with the mitochondrial fission protein DRP1. Previous reports have shown that LRRK2 can physically interact with DRP1 and that LRRK2 mediates mitochondrial fragmentation through DRP1 (*Wang et al., 2012*). Overexpression of both wild type LRRK2 and the PD-associated G2019S allele of LRRK2 have been shown to cause mitochondrial fragmentation (*Wang et al., 2012*). Curiously, we observe a similar phenotype in macrophages lacking LRRK2 (*Figure 3C*). Previous studies have linked LRRK2 and LRRK2 kinase activity to DRP1 activation via phosphorylation at several sites including T595 in neurons (*Su and Qi, 2013*) and S616 in a neuron-like carcinoma cell line (*Esteves et al., 2015*). Our observation that *Lrrk2* KO macrophages accumulate phospho-S616 DRP1 and exhibit increased fragmentation of the mitochondrial network indicates that LRRK2 is not required for DRP1 phosphorylation or activation of mitochondrial fission in macrophages. Indeed, other kinases have been shown to phosphorylate DRP1 at S616 in other cell types, including ERK2 (*Kashatus et al., 2015*) and CDK1 (*Taguchi et al., 2007*). We propose that loss of LRRK2 could alter DRP1 phosphorylation indirectly through changing serine accessibility or protein-protein interactions, or by modifying other pathways that control mitochondrial turnover or lysosome homeostasis. It will be important for future studies to compare the molecular mechanisms driving the DRP1-dependent mitochondrial fission defects in *Lrrk2* KO cells and in cells harboring the PD-associated 'gain of function' G2019S allele.

Mtb is a potent activator of cytosolic DNA sensing (*Manzanillo et al., 2012*; *Watson et al., 2012*), and type I IFNs are important biomarkers of Mtb infection associated with poor outcomes in humans and in mouse models of infection (*Berry et al., 2010*). New insights into the requirement of IFNAR signaling for nitric oxide production in macrophages ex vivo suggest critical roles for type I IFN induction in cell-intrinsic control of Mtb replication (*Banks et al., 2019*). However, the degree to which these macrophage phenotypes translate to mouse models of infection remains poorly understood. Although we observed a striking type I IFN defect (both higher basal levels and blunted induction) in a number of macrophage primary cells and cell lines, we did not find major differences in infection outcomes in *Lrrk2* HET vs. KO mice. Our previous experiments demonstrated that while

loss of cGAS almost completely abrogates type I IFN expression in macrophages, it has only minor effects in vivo (serum IFN-β levels and lung type I IFN/ISG expression levels) (*Collins et al., 2015*; *Watson et al., 2015*), suggesting that Mtb infection can elicit type I IFN expression in important cGAS-independent ways in vivo that we do not yet fully understand. Another recent publication that investigated the role of LRRK2 in controlling Mtb infection does report a significant decrease in CFUs in *Lrrk2* KO mice at very early infection time points (Day 7 and 14), which correlates with increased inflammation in the lungs (as we also report) (*Härtlova et al., 2018*). It is likely that minor discrepancies between our data and that reported by Härtlova et al. are the consequence of differences in mouse and Mtb strains and the fact that we compared *Lrrk2* KO and HET littermate controls as opposed to WT controls. It will be crucial moving forward to more directly interrogate the molecular drivers of inflammation and Mtb pathogenesis in *Lrrk2* KO mice as well as in mouse genotypes associated with human disease susceptibility, for example *LRRK2* G2019S. Because LRRK2 inhibitors are a major area of drug development for the treatment of PD, it is crucial to understand how both loss of and mutations in this protein might impact the ability of patients receiving such therapies to respond to and clear infection.

# Materials and methods

**Key resources table**

| Reagent type (species) or resource | Designation | Source or reference | Identifiers | Additional information |
|---|---|---|---|---|
| Gene (*Lrrk2*) | *Lrrk2*; LRRK2 | NA | MGI:1913975 | |
| Genetic reagent (*Mycobacterium tuberculosis*) | Mtb; Erdman | *Watson et al., 2015*; *Watson et al., 2012* | | |
| Genetic reagent (*Mycobacterium leprae*) | Mlep | National Hansen's Disease Program | | |
| Genetic reagent (*Mus musculus*) | *Lrrk2* KO; C57BL/6-*Lrrk2*tm1.1Mjff/J | Jackson Labs | 16121 Lrrk2 KO | |
| Genetic reagent (*Mus musculus*) | *Ifnar* KO; B6(Cg)-Ifnar1tm1.2Ees/J | Jackson Labs | 28288 Ifnar1 KO | |
| Genetic reagent (*Mus musculus*) | *Tfam* HET | obtained from A. P. West TAMHSC | Tfam HET | DOI: 10.1016/j.ajpath.2011.10.003 |
| Genetic reagent (*Mus musculus*) | *Cgas* KO; B6(C)-Cgastm1d (EUCOMM)Hmgu/J | obtained from A. P. West TAMHSC | cGAS KO | |
| Cell line (*Mus musculus*) | RAW 264.7 *Lrrk2* Parental; WT | ATCC | ATCC SC-6003 | |
| Cell line (*Mus musculus*) | RAW 264.7 *Lrrk2* KO | ATCC | ATCC SC-6004 | |
| Cell line (*Mus musculus*) | RAW 264.7 | ATCC | ATCC TIB-71 | Cell line maintained in the Watson lab |
| Cell line (*Homo sapiens*) | U937 | ATCC | ATCC CRL-1593.2 | Cell line maintained in the Watson lab |
| Antibody | anti-pSTAT1 Rabbit monoclonal | Cell Signaling | (Tyr701) (58D6) #9167 | (1:1000 WB) |
| Antibody | anti-STAT1 Rabbit monoclonal | Cell Signaling | (D4Y6Z) #14995 | (1:1000 WB) |
| Antibody | anti-pIRF3 Rabbit monoclonal | Cell Signaling | (Ser396) (4D4G) #4947 | (1:1000 WB) |
| Antibody | anti-IRF3 Rabbit monoclonal | Cell Signaling | (D83B9) #4302 | (1:1000 WB) |
| Antibody | anti-Beta tubulin Rabbit polyclonal | Abcam | ab15568 | (1:5000 WB) |

*Continued on next page*

*Continued*

| Reagent type (species) or resource | Designation | Source or reference | Identifiers | Additional information |
|---|---|---|---|---|
| Antibody | anti-pDRP1 Rabbit polyclonal | Cell Signaling | (Ser616) #3455 | (1:75 FC), (1:1000 WB) |
| Antibody | anti-DRP1 Rabbit monoclonal | Abcam | ab184247 | (1:1000 WB) (1:200 IF) |
| Antibody | anti-IFNB Rabbit polyclonal | PBL Assay Science | 32400–1 | (1:250 neutralizing) |
| Antibody | anti-PFAS Rabbit polyclonal | Bethyl Laboratories | A304-219A | (1:200 IF) |
| Antibody | Anti-TOM20 Mouse monoclonal | Millipore, via A.P. West lab TAMHSC | MABT166 | (1:200 IF) |
| Antibody | anti-TFAM Rabbit polyclonal | Millipore, via A.P. West lab TAMHSC | ABE483 | (1:1000 WB) |
| Antibody | anti-VDAC Rabbit polyclonal | Protein Tech, via A.P. West lab TAMHSC | 55259–1-AP | (1:1000 WB) |
| Antibody | Goat anti Rabbit IgG | Licor | IRDye 800CW | (1:10000 WB) |
| Antibody | Goat anti Rabbit IgG | Licor | IRDye 680CW | (1:10000 WB) |
| Antibody | Goat anti Rabbit IgG AF 488 | Invitrogen | A32731 | (1:500 FC) |
| Sequence-based reagent | interferon stimulatory DNA; ISD | IDT | annealed in house | (1 ug/ml) |
| Sequence-based reagent | CpG 2395 | Invivogen | tlrl-2395 | (1 uM) |
| Peptide, recombinant protein | recombinant IFNB | PBL Assay Science | 12405–1 | (200 IU/mL) |
| Commercial assay or kit | Seahorse XF mito stress kit | Agilent | 103708–100 | |
| Commercial assay or kit | Direct-zol RNA mini prep | Zymo research | R2052 | |
| Commercial assay or kit | Mouse Cytokine Array | Eve Technologies | MD13 panel | |
| Chemical compound, drug | thioglycollate | Fisher Scientific | BD 211716 | |
| Chemical compound, drug | DMXAA | Invivogen | tlrl-dmx | (5 ug/ml) |
| Chemical compound, drug | urate | Sigma Aldrich | U2625 | |
| Chemical compound, drug | IMP | Sigma Aldrich | 57510 | |
| Chemical compound, drug | hypoxanthine | Sigma Aldrich | H9377 | |
| Chemical compound, drug | FCCP | Sigma Aldrich | C2920 | (50 uM) |
| Chemical compound, drug | CLO97 | Invivogen | tlrl-c97 | (100 ng/ml) |
| Chemical compound, drug | JC1 dye | Thermofisher | T3168 | (1 uM) |
| Chemical compound, drug | TMRE | Thermofisher | T669 | (25 ng/ml) |
| Chemical compound, drug | ATP | Invivogen | tlrl-atpl | (5 uM) |

*Continued on next page*

*Continued*

| Reagent type (species) or resource | Designation | Source or reference | Identifiers | Additional information |
|---|---|---|---|---|
| Chemical compound, drug | rotenone | Sigma Aldrich | R8875 | (2.5 uM) |
| Chemical compound, drug | mitoTEMPO | Santa Cruz Biotechnology | (CAS 1569257-94-8) | (200 uM) |

## Primary cell culture

Bone marrow derived macrophages (BMDMs) were differentiated from bone marrow cells isolated by washing mouse femurs with 10 ml DMEM. Cells were then centrifuged for 5 min at 1000 rpm and resuspended in BMDM media (DMEM, 20% FBS, 1 mM Sodium pyruvate, 10% MCSF conditioned media). BM cells were counted and plated at $5 \times 10^6$ in 15 cm non-TC treated dishes in 30 ml complete media and fed with an additional 15 ml of media on Day 3. On Day 7, cells were harvested with 1x PBS-EDTA.

Mouse embryonic fibroblasts (MEFs) were isolated from embryos. Briefly, embryos were dissected from yolk sacs, washed two times with cold 1x PBS, decapitated, and peritoneal contents were removed. Headless embryos were disaggregated in cold 0.05% trypsin-EDTA and incubated on ice for 20 min, followed by incubation at 37°C for an additional 20 min and DNase treatment (20 min, 37°C, 100 ug/ml). Supernatants were removed and spun down at 1000 rpm for 5 min. Cells were resuspended in DMEM, 10% FBS, 1 mM sodium pyruvate, and plated in 15 cm TC treated dishes, three dishes per embryo. MEFs were allowed to expand for 2–3 days before harvest with Trypsin 0.05% EDTA.

Peritoneal macrophages (PEMs), were elicited by intraperitoneal injection of 1 ml 3% Thioglycollate broth (BD Biosciences) for 4 days prior to harvest. For harvest, PEMs were isolated from mice by lavage (1x PBS 4°C) and resuspended in RPMI 1640 media with 20% FBS, 1 mM sodium pyruvate and 2 mM L-Glutamine. Following overnight incubation at 37°C, cells were washed twice (1x PBS 37°C) to remove non-adherent cells.

## Cell lines, treatments, and stimulations

RAW 264.7 and U937 cell lines were each purchased from ATCC. All our cell lines are minimally passaged to maintain genomic integrity and new cell lines are generated from these low passage stocks. Cell lines were passaged no more than 10 times. Our cell lines stocks have all tested negative for mycoplasma contamination. RAW 264.7 *Lrrk2* KO cells (ATCC SC-6004) generated by the MJFF, were obtained from the ATCC and used with wild type control *Lrrk2* parental RAW 264.7 (ATCC SC-6003). To deplete mtDNA, RAW 264.7 cells were seeded at $2 \times 10^6$ cells/well in 10 cm non-TC treated dishes and cultured for 4 days in complete media (DMEM, 10% FBS, 1 mM sodium pyruvate) with 10 µM ddC. Cells were split and harvested with 1x PBS-EDTA.

Prior to treatment/stimulation, BMDMs were plated in 12 well plates at $5 \times 10^5$ cells/well, or 6-well plates at $1 \times 10^6$ cells/well. MEFs were plated in 12 well dishes at $3 \times 10^5$ cells/well. PEMs were plated in 24-well flat-bottomed plates at $1 \times 10^6$ cells/ well. RAW 264.7 cells were plated at $7.5 \times 10^5$ cells/well. Cells were stimulated for 4 hr with 1 µM CLO97, 100 ng/ml LPS, or transfected 1 µg/ml ISD, 1 µg/ml poly(I:C), 1 µg/ml cGAMP with lipofectamine. Cells were transfected for 4 hr with 1 µM CpG 2395 with Gene Juice. Cells were stimulated for 2–4 hr with 10 µM DMXAA (RAW 264.7) or 200 IU IFN-β (BMDMs).

## Mice

*Lrrk2* KO mice (C57BL/6-Lrrk2tm1.1Mjff/J) stock #016121, and *Ifnar* KO mice (B6(Cg)-Ifnar1tm1.2Ees/J) stock #028288 were purchased from Jackson Laboratories (Bar Harbor, ME). *Tfam* HET (*An et al., 2008*; *Schendel et al., 1988*; *Zhao et al., 2013*) and *Mb21d1* (*cGas*) KO (B6(C)-Cgastm1d (EUCOMM)Hmgu/J) mice were provided by A. Phillip West at Texas A&M Health Science Center. *Lrrk2* KO mice used in experiments were backcrossed once onto C57BL6/NJ by the Jackson labs and then maintained with filial breeding. (N1F8). The *Lrrk2* KO strain has been maintained with filial breeding on a C57BL6/NJ background for five more generations. When breeding *Lrrk2* KO mice to *Ifnar* KO, *cGas* KO and *Tfam* HET strains all of which are on a C57BL6/J background, mice were

backcrossed for two generations onto the C57BL6/NJ and then were maintained with filial breeding (currently F3). All mice used in experiments were compared to age- and sex- matched controls. In order to ensure littermate controls were used in all experiments *Lrrk2* KO crosses were made with (KO) *Lrrk2*$^{-/-}$ x (HET) *Lrrk2*$^{+/-}$ mice. Mice used to generate BMDMs and PEMs were between 8 and 12 weeks old. Mice were infected with Mtb at 10 weeks. Embryos used to make primary MEFs were 14.5 days post coitum. All animals were housed, bred, and studied at Texas A&M Health Science Center under approved Institutional Care and Use Committee guidelines.

### *Mycobacterium* macrophage infections

The Erdman strain was used for all Mtb infections (*Watson et al., 2015*; *Watson et al., 2012*). Low passage lab stocks were thawed for each experiment to ensure virulence was preserved. Mtb was cultured in roller bottles at 37°C in Middlebrook 7H9 broth (BD Biosciences) supplemented with 10% OADC, 0.5% glycerol, and 0.1% Tween-80 or on 7H11 plates. All work with Mtb was performed under Biosafety Level 3 (BSL3) containment using procedures approved by the Texas A and M University Institutional Biosafety Committee. Prior to infection, BMDMs were seeded at $1.2 \times 10^6$ cells/well (6-well dish) or $3 \times 10^5$ cells/well (12-well dish), RAW cells at $5 \times 10^5$ cells/well (12-well dish), and U937s at $1 \times 10^6$ cells/well. U937s were cultured with 10 ng/ml phorbol 12-myristate 13-acetate (PMA) for 48 hr to induce differentiation and then recovered in fresh media for an addition 24 hr prior to infection. To prepare the inoculum, bacteria grown to log phase (OD 0.6–0.8) were spun at low speed (500 g) to remove clumps and then pelleted and washed with 1x PBS twice. Resuspended bacteria were briefly sonicated and spun at low speed once again to further remove clumps. The bacteria were diluted in DMEM + 10% horse serum and added to cells, MOI = 10. Cells were spun with bacteria for 10 min at 1000 g to synchronize infection, washed twice with PBS, and then incubated in fresh media. RNA was harvested from infected cells using 0.5–1.0 ml Trizol reagent 4 hr post-infection unless otherwise indicated.

*M. leprae* was cultivated in the footpads of nude mice and generously provided by the National Hansen's Disease Program. Bacilli were recovered overnight at 33°C, mixed to disperse clumps and resuspended in DMEM + 10% horse serum. Cells were infected as with Mtb but with an MOI of 50.

### Mtb mouse infections

All infections were performed using procedures approved by Texas A&M University Institutional Care and Use Committee. The Mtb inoculum was prepared as described above. Age- and sex-matched mice were infected via inhalation exposure using a Madison chamber (Glas-Col) calibrated to introduce 100–200 CFUs per mouse. For each infection, approximately five mice were euthanized immediately, and their lungs were homogenized and plated to verify an accurate inoculum. Infected mice were housed under BSL3 containment and monitored daily by lab members and veterinary staff. At the indicated time points, mice were euthanized, and tissue samples were collected. Blood was collected in serum collection tubes, allowed to clot for 1–2 hr at room temperature, and spun to separate serum. Serum cytokine analysis was performed by Eve Technologies. Organs were divided to maximize infection readouts (CFUs: left lobe lung and ½ spleen; histology: two right lung lobes and ¼ spleen; RNA: one right lung lobe and ¼ spleen). For histological analysis organs were fixed for 24 hr in either neutral buffered formalin and moved to ethanol (lung, spleen). Organs were further processed as described below. For cytokine transcript analysis, organs were homogenized in Trizol Reagent, and RNA was isolated as described below. For CFU enumeration, organs were homogenized in 5 ml PBS + 0.1% Tween-80, and serial dilutions were plated on 7H11 plates. Colonies were counted after plates were incubated at 37°C for 3 weeks.

### Histopathology

Lungs and spleens were fixed with paraformaldehyde, subjected to routine processing, embedded in paraffin, and 5 μm sections were cut and stained with hematoxylin and eosin (H and E) or acid-fast stain. A boarded veterinary pathologist performed a masked evaluation of lung sections for inflammation using a scoring system: score 0, none; score 1, up to 25% of fields; score 2, 26–50% of fields; score 3, 51–75% of fields; score 4, 76–100% of fields. To quantify the percentage of lung fields occupied by inflammatory nodules, scanned images of at least 2 sections of each lung were analyzed using Fiji Image J (*Johansen et al., 2009*) to determine the total cross-sectional area of inflammatory

nodules per total lung cross sectional area. Total neutrophil scores were determined using digital images of H and E slides divided into 500 × 500 um grids and counting the percentage of squares containing neutrophils (total PMN) or degenerate neutrophils.

## mRNA sequencing

RNA-seq represents analysis of 16 samples (biological quadruplicates of *Lrrk2* HET uninfected, *Lrrk2* HET +Mtb, *Lrrk2* KO uninfected, and *Lrrk2* KO +Mtb; one sample from the *Lrrk2* HET +Mtb group was removed from the analysis due to poor quality sequencing). Briefly, RNA was isolated from BMDMs using PureLink RNA mini kits (Ambion) and quantified on an Agilent Bioanalyzer 2100. PolyA+ PE 100 libraries were sequenced on a HiSeq 4000 at the UC Davis Genome Center DNA Technologies and Expression Analysis Core. Raw reads were processed with expHTS (*Streett et al., 2015*) to trim low-quality sequences and adapter contamination, and to remove PCR duplicates. Trimmed reads for each sample were mapped to the *Mus musculus* Reference genome (RefSeq) using CLC Genomics Workbench 8.0.1. Relative transcript expression was calculated by counting Reads Per Kilobase of exon model per Million mapped reads (RPKM). Differential expression analyses were performed using CLC Genomics Workbench EDGE test. Differentially expressed genes were selected as those with p-value threshold <0.05 in the heatmaps represented. Heatmaps were generated using GraphPad Prism software (GraphPad, San Diego, CA).

## qRT-PCR

RNA was isolated using Direct-zol RNAeasy kits (Zymogen). cDNA was synthesized with BioRad iScript Direct Synthesis kits (BioRad) per manufacturer's protocol. qRT-PCR was performed in triplicate wells using PowerUp SYBR Green Master Mix. Data were analyzed on a QuantStudio 6 Real-Time PCR System (Applied Biosystems).

## Cytosolic DNA isolation

$3 \times 10^6$ MEFs or $1 \times 10^7$ RAW 264.7 cells were plated in 10 cm dishes. The next day, confluent plates were treated as indicated with inhibitors. To harvest, cells were lifted with 1x PBS-EDTA. To determine total DNA content, 1% of the input was saved and processed by adding NaOH to 50 mM, boiling 30 min, and neutralizing with 1:10 1M Tris pH 8.0. To isolate cytosolic DNA, the cells were pelleted and resuspended in digitonin lysis buffer (150 mM HEPES pH 7.4, 50 mM NaCl, 10 mM EDTA, 25–50 µg/ml digitonin). Cells were incubated for 15 min at 4°C on an end-over-end rotator. Cells were spun at 980 x g for 3 min, and the supernatant was collected and spun again at 15,000 x g for 3 min to remove any remaining organelle fragments. DNA from the cleared supernatant (cytosolic fraction) was then extracted via phenol:chloroform (1:1 supernatant:phenol/chloroform). The DNA from the aqueous layer was precipitated in 0.3 M sodium acetate, 10 mM magnesium chloride, 1 µg/ml glycogen, and 75% ethanol. After freezing overnight at −20°C, the DNA was pelleted, washed in 70% ethanol, dried, resuspended in TE, and solubilized at 50°C for 30 min. qPCR was performed on the input (1:50 dilution) and cytosolic (1:2 dilution) samples using nuclear (*Tert*) and mitochondrial (*16*s and *cytB*) genes. The total and cytosolic mitochondrial DNA was normalized to nuclear DNA in order to control for variation in cell number.

## Western blot

Cells were washed with PBS and lysed in 1x RIPA buffer with protease and phosphatase inhibitors, with the addition of 1 U/ml Benzonase to degrade genomic DNA. Proteins were separated by SDS-PAGE and transferred to nitrocellulose membranes. Membranes were blocked for 1 hr at RT in LiCOR Odyssey blocking buffer. Blots were (Licor) and incubated overnight at RT with the following antibodies: IRF3 (Cell Signaling, 1:1000); pIRF3 Ser396 (Cell Signaling, 1:1000); STAT1 (Cell Signaling, 1:1000); pSTAT1 Tyr701 (Cell Signaling, 1:1000); Beta Actin (Abcam, 1:2000), β-Tubulin (Abcam, 1:5000); DRP1 (Cell Signaling, 1:1000); pDRP1 Ser616 (Cell Signaling, 1:1000), TFAM (Millipore, 1:1000), VDAC (Protein Tech, 1:1000). Membranes were incubated with appropriate secondary antibodies for 2 hr at RT prior to imaging on a LiCOR Odyssey Fc Dual-Mode Imaging System.

## Seahorse metabolic assays

Seahorse XF mito stress test kits and cartridges were prepared per manufacturers protocol as described in *An et al. (2008)*; *Schendel et al. (1988)*; *Zhao et al. (2013)* and analyzed on an Agilent Seahorse XF 96-well Analyzer. BMDMs were seeded at $5 \times 10^4$ cells/well overnight and treated with 200 µM mitoTEMPO, IFN-β neutralizing Ab, or sodium pyruvate at 0, 1, or 2 mM final concentration.

## Immuno-multiplex assay

Sera was analyzed by Eve Technologies: Mouse Cytokine Array/Chemokine Array 13-plex Secondary Panel (MD13). Briefly, sera was isolated following decapitation in Microtainer serum separator tubes (BD Biosciences) followed by 2x sterile filtration with Ultrafree-MC sterile filters, 10 min at 10,000 rpm (Millipore Sigma). For analysis sera was prediluted 1:1 to a final volume of 100 µl in 1x PBS pH 7.4 and assayed/analyzed in duplicate.

## Immunofluorescence microscopy

MEFs were seeded at $1 \times 10^5$ cells/well on glass coverslips in 24-well dishes. Cells were fixed in 4% paraformaldehyde for 10 min at RT and then washed three times with PBS. Coverslips were incubated in primary antibody diluted in PBS + 5% non-fat milk + 0.1% Triton-X (PBS-MT) for 3 hr. Cells were then washed three times in PBS and incubated in secondary antibodies and DAPI diluted in PBS-MT for 1 hr. Coverslips were washed twice with PBS and twice with deionized water and mounted on glass slides using Prolong Gold Antifade Reagent (Invitrogen).

## Flow cytometry

### JC-1 assay to assess mitochondrial membrane potential

Cells were released from culture plates with 1x PBS + EDTA (BMDMs, RAW 264.7) or Accutase (MEFs). Single cell suspensions were made in 1x PBS 4% FBS. JC-1 dye was sonicated for 5 min with 30 s intervals. Cells were stained for 30 min at 37°C in 1 µM JC-1 dye and analyzed on an LSR Fortessa X20 (BD Biosciences). Aggregates were measured under Texas Red (610/20 600LP) and monomers under FITC (525/50 505LP). To assess mitochondrial membrane potential under stress, cells were treated for 3 hr with 2.5 µM rotenone prior to being lifted of the culture plates. 5 µM ATP was then added for 5, 15, or 30 min, or 50 µM FCCP was added for 15 min. For rescue assays, cells were treated for 4 hr with Mdivi-1 at 10 µM or 50 µM or overnight with 200 µM mitoTEMPO or 100 µM urate (Sigma Aldrich).

### TMRE assay to assess mitochondrial membrane potential

Cells were released from culture plates with 1x PBS + EDTA (BMDMs, RAW 264.7) or Accutase (MEFs). Single cell suspensions were made in 1x PBS 4% FBS. Cells were stained for 20 min at 37°C in 25 nM TMRE dye and analyzed on an LSR Fortessa X20 (BD Biosciences). Fluorescence was measured under PE (585/15). To assess mitochondrial membrane potential under stress, cells were treated for 3 hr with 2.5 µM rotenone prior to being lifted of the culture plates. 5 µM ATP or 50 µM FCCP was then added for 15 min. For rescue assays, cells were treated for 4 hr with 50 µM Mdivi-1.

### Phospho-DRP1 assay

Cells were washed once in 1x PBS and fixed in 4% cold PFA for 10 min. Cells were then permeabilized with 0.3% Triton-X for 15 min, followed by 30 min block in 0.1% Triton-X + 5% normal rat serum. Cells were incubated in pDRP1 Ser616 antibody (Cell Signaling) overnight at 4°C in 0.1% Triton-X + 1% BSA and then in secondary antibodies (AF488 Goat anti-Rabbit). Cells were analyzed on an LSR Fortessa X20 (BD Biosciences) under FITC (525/50 505LP). For rescue and exacerbation assays, cells were treated with 100 µM $H_2O_2$ for 1 hr at 37°C or with 50 µM Mdivi-1 for 12 hr.

## LC-MS/MS

### Sample extraction

Samples were weighed and extracted with a methanol:chloroform:water-based extraction method. Briefly 800 µl ice cold methanol:chloroform (1:1, v:v) was added to samples in a bead-based lysis tube. Samples were extracted on a Precyllys 24 tissue homogenizer for 30 s at a speed of 6000 rpm.

The supernatant was collected, and samples were homogenized a second time with 800 µl ice methanol:chloroform. 600 µl ice cold water was added to the combined extract, vortexed, and centrifuged to separate the phases. The upper aqueous layer was passed through a 0.2 µm nylon filter. 500 µl of the filtered aqueous phase was then passed through a 3 kDa cutoff column and the flow through was collected for analysis.

### Sample analysis

Untargeted liquid chromatography high resolution accurate mass spectrometry (LC-HRAM) analysis was performed on a Q Exactive Plus Orbitrap mass spectrometer coupled to a binary pump HPLC (UltiMate 3000, Thermo Scientific). For acquisition, the Sheath, Aux and Sweep gasses were set at 50, 15 and 1 respectively. The spray voltage was set to 3.5 kV (Pos) or 2.8 kV (Neg) and the S-lens RF was set to 50. The source and capillary temperatures were both maintained at 350°C and 350°C, respectively. Full MS spectra were obtained at 70,000 resolution (200 m/z) with a scan range of 50–750 m/z. Full MS followed by ddMS2 scans were obtained at 35,000 resolution (MS1) and 17,500 resolution (MS2) with a 1.5 m/z isolation window and a stepped NCE (20, 40, 60). Samples were maintained at 4°C before injection. The injection volume was 10 µl. Chromatographic separation was achieved on a Synergi Fusion 4 µm, 150 mm x 2 mm reverse phase column (Phenomenex) maintained at 30°C using a solvent gradient method. Solvent A was water (0.1% formic acid). Solvent B was methanol (0.1% formic acid). The gradient method used was 0–5 min (10% B to 40% B), 5–7 min (40% B to 95% B), 7–9 min (95% B), 9–9.1 min (95% B to 10% B), 9.1–13 min (10% B). The flow rate was 0.4 mL/min. Sample acquisition was performed Xcalibur (Thermo Scientific). Data analysis was performed with Compound Discoverer 2.1 (Thermo Scientific).

Inosine 5′-monophosphate disodium salt hydrate (57510; Sigma-Aldrich) and hypoxanthine (H9377; Sigma-Aldrich) pure molecular weight standards were used to verify the retention time and mass spectra of the unknown compounds.

## Statistical analysis

All data are representative of two or more independent experiments with n = 3 or greater. In the majority of experiments involving induction or infection, the independent variables were heavily skewed (i.e. the variable of Mtb infection vs. variable of genotype). Therefore to avoid a type II error, a log transformation was performed on the data prior to analysis. Data were then analyzed by two-way, or three-way ANOVA, followed by Tukey's post-hoc test to determine significance. Experiments involving one independent variable were analyzed without a log transformation. Here significance was determined using a Student's two-tailed T test or a one-way ANOVA followed by a Tukey's multiple comparisons test. For LCM/MS MS significance was determined with a one-way ANOVA followed by a Tukey HSD post-hoc test. A Benjamini-Hochberg correction was used for the false discovery rate. Graphs were generated using Prism (GraphPad). In order to depict baseline and induced gene expression on the same graph, we have broken the y-axis into segments where needed. Error bars represent SEM.

For mouse experiments, we estimated that detecting a significant effect requires two samples to differ in CFUs by 0.7e̊10. Using a standard deviation of 0.35e̊10 for each population, we calculated that a minimum size of 5 age- and sex-matched mice per group per time point is necessary to detect a statistically significant difference by a t-test with alpha (2-sided) set at 0.05 and a power of 80%. Therefore, we used a minimum of 5 mice per genotype per time point to assess infection-related readouts. For statistical comparison, each experimental group was tested for normal distribution. Data were tested using a Mann-Whitney test.

## Acknowledgements

We'd like to thank Cory Klemashevich at the TAMU Integrated Metabolomics Analysis Core for his help with the metabolomics analysis. We would also like the acknowledge Monica Britton at the University of California, Davis DNA Technologies and Expression Analysis Core Library for her help with the RNA-seq analysis. We would like to acknowledge the members of the Patrick and Watson labs for their helpful discussions and feedback as well as Elizabeth Case for her help proofreading. We'd like to thank our collaborators A Phillip West and the West lab for their help with mitochondrial experiments and for providing us with *Tfam* HET and *Mb21d1* (*cGas*) KO mice as well as Rahul

Srinivasan and Taylor Huntington for help with experiments related to LRRK2. We'd lastly like to thank Nevan Krogan at University of California, San Francisco for his help with the conceptual design of this manuscript. This work was supported by funds from the Michael J Fox Foundation for Parkinson's Research, grant 12185 (to ROW), the National Institutes of Health, grant 1R01AI125512 (to ROW) and R35GM133720 (to KLP), and the Texas A&M Clinical Science and Translational Research (CSTR) Pilot Grant Program (to ROW). Additional funding was provided by the Parkinson's Foundation Postdoctoral Fellowship (to CGW), NIH training grant 5T32OD011083-10 and the Texas A&M CVM Postdoctoral Trainee Research Training Grant (to KJV).

## Additional information

### Funding

| Funder | Grant reference number | Author |
|---|---|---|
| Michael J. Fox Foundation for Parkinson's Research | M1801235 | Robert O Watson |
| National Institute of Allergy and Infectious Diseases | R21AI40004 | Robert O Watson |
| National Institute of General Medical Sciences | R35GM133720 | Kristin L Patrick |
| Parkinson's Disease Foundation | | Chi G Weindel |
| National Institutes of Health | 5T32OD011083-10 | Krystal J Vail |
| Michael J. Fox Foundation for Parkinson's Research | 12185 | Robert O Watson |
| National Institute of Allergy and Infectious Diseases | 1R01AI12551 | Robert O Watson |

The funders had no role in study design, data collection and interpretation, or the decision to submit the work for publication.

### Author contributions

Chi G Weindel, Conceptualization, Formal analysis, Funding acquisition, Investigation, Visualization, Methodology; Samantha L Bell, Conceptualization, Formal analysis, Investigation, Visualization, Methodology; Krystal J Vail, Formal analysis, Funding acquisition, Investigation; Kelsi O West, Formal analysis, Investigation, Visualization; Kristin L Patrick, Conceptualization, Formal analysis, Supervision, Funding acquisition, Methodology; Robert O Watson, Conceptualization, Formal analysis, Supervision, Funding acquisition, Investigation, Visualization, Methodology

### Author ORCIDs

Samantha L Bell (iD) https://orcid.org/0000-0002-5453-3203
Krystal J Vail (iD) http://orcid.org/0000-0002-1964-7985
Kristin L Patrick (iD) http://orcid.org/0000-0003-2442-4679
Robert O Watson (iD) https://orcid.org/0000-0003-4976-0759

### Ethics

Animal experimentation: This study followed the recommendations in the Guide for the Care and Use of Laboratory Animals by the National Research Council. All animals were housed, bred, and studied at Texas A&M Health Science Center using protocols reviewed and approved by the institutional animal care and use committee (IACUC) of Texas A&M University (protocol #2018-0125).

### Decision letter and Author response

Decision letter https://doi.org/10.7554/eLife.51071.sa1
Author response https://doi.org/10.7554/eLife.51071.sa2

## Additional files

### Supplementary files

• Supplementary file 1. RNA-seq analysis of *Lrrk2* KO and HET BMDMs. Fold change and p-values separated into tabs. Tab 1: *Lrrk2* HET vs. KO uninfected all genes fold change and p-value. Tab 2: Fold change values for genes shown in *Figure 1A* heatmap. All p-values<0.05. Tab 3: *Lrrk2* HET vs. KO +Mth (4 hr time point) all genes fold change and p-value. Tab 4: Fold change values for genes shown in *Figure 1C* heatmap. All p-values<0.05. Tab 5: Fold change and p-values for all pairwise comparisons of RNA-seq data. Tab 6: Fold change and p-values for genes shown in *Figure 1—figure supplement 1A* heatmap.

• Supplementary file 2. LC/MS/MS metabolite identification. Tab 1: All compounds identified and their relative amounts (group area) for each genotype (*Lrrk2* KO and HET). Tab 2: Data for metabolites highlighted in the manuscript (IMP, hypoxanthine, oxidized glutathione).

• Supplementary file 3. Summary of statistical analyses. Statistical test, n, and p-value for each data point in manuscript. Tabs labeled according to Figure.

• Transparent reporting form

### Data availability

All data generated or analyzed during this study are included in the manuscript and supporting files. Source data files have been provided for Figures 1 and 5.

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
