## [Decision Letter]

**Acceptance summary:**

This paper demonstrates that macrophages lacking the Parkinson's Disease-associated gene LRRK2 exhibit an altered type I interferon response characterized by high basal levels of type I IFNs and a blunted ability to induce type I IFN expression in response to stimulation. The high basal levels of type I IFNs are due to alterations in mitochondrial homeostasis and cytosolic leakage of mitochondrial DNA, which engages the cGAS-STING DNA-sensing immune pathway. The mitochondrial and immune dysfunction in LRRK2 deficient animals may be important to consider to explain the role of LRRK2 in Parkinson's and infectious diseases such as tuberculosis.

**Decision letter after peer review:**

Thank you for submitting your article "LRRK2 regulates innate immune responses and neuroinflammation during *Mycobacterium tuberculosis* infection" for consideration by *eLife*. Your article has been reviewed by three peer reviewers, including Russell E Vance as the Reviewing Editor and Reviewer #3, and the evaluation has been overseen by Tadatsugu Taniguchi as the Senior Editor.

Overall we are enthusiastic about the potential importance of the work. However, some significant revisions are required. The reviewers have discussed the reviews with one another and the Reviewing Editor has drafted this consolidated decision to help you prepare a revised submission.

Summary:

This manuscript addresses the immunological phenotype of mice lacking LRRK2, a poorly understood kinase with proposed roles in resistance to infection and in Parkinson's Disease. The authors provide evidence that loss of LRRK2 results in mitochondrial dysfunction and chronic activation of the cytosolic innate immune cGAS-STING pathway. These results have important implications in understanding the complex pathology seen in LRRK2 deficiency.

Essential revisions:

Overall: Significant questions were raised about the analysis and statistics in the first five figures. Nevertheless we consider the core observations in these figures to be potentially important. The last two figures of the paper attempts to connect LRRK2 deficiency and neuroinflammation during infection with *Mycobacterium tuberculosis*. For reasons outlined below, we found this part of the paper considerably less convincing. We feel the authors are overstraining to push an interpretation that is not well justified--a strategy that may be necessary for other journals, but not *eLife*. Overall, we feel that a reworking of the manuscript as suggested below would potentially allow for publication in *eLife* without a need for much substantial additional experimentation.

1) The reviewers suggest refocusing the manuscript, including the title and Abstract, on the important observation that loss of LRRK2 leads to chronic cGAS-STING activation and decreased responsiveness to IFN inducing stimuli (Figures 1-5). We suggest removing the data that attempt to establish a role for LRRK2 in Mtb infection and neuroinflammation (Figures 6,7). Our rationale is that the in vivo infection/neuroinflammation phenotypes are either absent or mild and equivocal (e.g., no role for LRRK2 in microglia reactivity, astrocyte reactivity up at one timepoint, down at another). In addition, there were concerns that the in vivo data were quantified/normalized appropriately. The in vitro microglia data (e.g., Figure 7F,H) appear to show the complete opposite of the macrophage data presented earlier in the manuscript (i.e., down instead of up in unstimulated, enhanced instead of dampened responses to stimulation). This is confusing and undermines the rest of the paper. Overall, the attempt to push the 2-hit model of neurodegeneration strikes as us forced overinterpretation. Indeed, no data on neuronal loss is shown. Moreover, the evidence that loss of LRRK2 contributes to PD risk is equivocal. Many assays, including direct assays of LRRK2 function, suggest that mutant alleles retain function. There is some direct human genetic evidence recently that loss of function alleles are present in humans but not pathogenic. If the authors wish to determine if pathogenic mutations associated with PD alter Mtb infection, they would have to use G2019S or R1441C knockin mice. Lastly, the phenotype of LRRK2 KO mice infected with Mtb has been previously reported (Hartlova et al) and the mild inflammatory phenotype seen by the present authors broadly resembles this previous report, though it is appreciated that there are some differences (e.g., no CFU or cytokine differences, unlike Hartlova). If the authors wish to 'correct the record', it might be appropriate to include the Mtb cfu/cytokine/neutrophil data (Figure S6A-C, 6A-B) as supplemental, along with a discussion of why you may have obtained different results (e.g., is this related to the use of +/- hets instead of +/+ WT mice?). However, we suggest 'saving' the neuroinflammation data for a subsequent paper when further work will allow some of the complex phenotypes to be better understood.

2) The reviewers have significant concerns about how the data were analyzed. We feel it is inappropriate, e.g., in Figure 1 but also elsewhere, to present the response of control and KO genotypes to infection (normalized to unstimulated) separately from the baseline responses of genotypes at baseline. Typically, one would compare treatment and genotype in one experiment and use a two-factor analysis. Unless there is a strong rationale for performing the experiments separately and in fact that is what was done, this figure needs reworking and p values for genotype, treatment and interaction terms generated. We feel it is not appropriate to express results as fold induction when the baseline measurements are not equivalent across conditions. This concern also affects Figures 2, 3, 4E, 5G. The authors may have chosen their convoluted presentation because the basal levels of gene expression are very low compared to the induced levels and plotting all four conditions on the same axis would minimize the differences in basal expression. This might be solvable by plotting the data on a log scale; or if that is unsatisfactory, by plotting all the data in a single graph but with an additional inset to highlight the basal differences. In general, the statistical reporting should be improved by stating which test was used in which figure, what n precisely means in each figure and please also give the value of the test statistic, particularly for parent ANOVAs where these were used. Related to the above, in panels 3B, 3I, 3J, 4E, 5G and possibly elsewhere, please indicate if the 2nd and 4th conditions are statistically different from each other (i.e., does mitoT or ddC or cGAS KO, or urate, etc., significantly enhance the ability of LRRK2 KO cells to respond to stimuli?). For the RNAseq data in Figure 1: it was very hard to understand what the n of samples is. It appears from 1A and 1C that this an n=1 experiment, based on the lack of p values in A and the single lane in the heatmaps. If so, this is simply not an appropriate experiment to report. If not, then the presentation needs to show individual values for replicates.

3) For Figure 4 and later in Figure 5, many studies have raised concerns about JC-1 because the red/green shift is dependent not just on membrane potential but also local concentration. Here, if the mitochondria show different volumes based on prior figures then this may contribute to some of the apparent effects. It is advised that JC-1 dye assays are run with CCCP controls to ensure the signal is sensitive to changes in mitochondrial membrane potential. Can the authors show this control data to ensure the signal changes are due to altered membrane potential. In addition, it is strongly recommended that these figures are confirmed with TMRM or TMRE, which are less prone to these artifacts.

4) Can the authors demonstrate alterations in DRP1 phosphorylation by loss-of-Lrrk2 by western blot? The authors have shown images of DRP1 puncta, however there is no quantification to ensure no changes in overall DRP1 expression. Again, this should be done via western blot. Furthermore the authors claim that DRP1 distribution is not different between genotypes but no quantification has been shown; by eye, it does indeed seem that there is decreased DRP1/TOM20 colocalisation in the Lrrk2-KO cells. As well, authors show histograms from flow data regarding DRP1-S616 levels, but this is not quantified anywhere. Furthermore, the authors report an increase in DRP1 phosphorylation via flow in Lrrk2-KO BMDCs. This is in contrast to other reports that show increased LRRK2 kinase activity increases DRP1 (doi: 10.5607/en.2018.27.3.171). Can the authors speculate how both increased kinase LRRK2 activity and loss of LRRK2 can induce the same phenotype?

---

## [Author Response]

Essential revisions:Overall: Significant questions were raised about the analysis and statistics in the first five figures. Nevertheless we consider the core observations in these figures to be potentially important. The last two figures of the paper attempts to connect LRRK2 deficiency and neuroinflammation during infection with Mycobacterium tuberculosis. For reasons outlined below, we found this part of the paper considerably less convincing. We feel the authors are overstraining to push an interpretation that is not well justified--a strategy that may be necessary for other journals, but not eLife. Overall, we feel that a reworking of the manuscript as suggested below would potentially allow for publication in eLife without a need for much substantial additional experimentation.1) The reviewers suggest refocusing the manuscript, including the title and Abstract, on the important observation that loss of LRRK2 leads to chronic cGAS-STING activation and decreased responsiveness to IFN inducing stimuli (Figures 1-5). We suggest removing the data that attempt to establish a role for LRRK2 in Mtb infection and neuroinflammation (Figures 6,7).

We thank the reviewers and editors for these constructive suggestions. We agree that the neuroinflammation portion of the story could be seen as underdeveloped mechanistically. As per your suggestion, we have removed Figures 6 and 7 from the manuscript and no longer make mention of any CNS phenotypes. We have refocused the manuscript to highlight our major findings that:

· Loss of LRRK2 impacts gene expression of type I IFN and ISGs in macrophages, resulting high basal levels and blunted induction levels

· High basal levels of type I IFNs in *Lrrk2* KO macrophages are driven by chronic cGAS signaling, resulting from DRP1-dependent mitochondrial damage and increased release of cytosolic mtDNA

· LRRK2’s contributions to purine metabolism and antioxidant production impact mitochondrial health and type I IFN expression in macrophages

· Mtb-infected *Lrrk2* KO mice experience exacerbated inflammation and lower expression of type I IFN transcripts in the lungs.

Our rationale is that the in vivo infection/neuroinflammation phenotypes are either absent or mild and equivocal (e.g., no role for LRRK2 in microglia reactivity, astrocyte reactivity up at one timepoint, down at another). In addition, there were concerns that the in vivo data were quantified/normalized appropriately. The in vitro microglia data (e.g., Figure 7F,H) appear to show the complete opposite of the macrophage data presented earlier in the manuscript (i.e., down instead of up in unstimulated, enhanced instead of dampened responses to stimulation). This is confusing and undermines the rest of the paper. Overall, the attempt to push the 2-hit model of neurodegeneration strikes as us forced overinterpretation. Indeed, no data on neuronal loss is shown. Moreover, the evidence that loss of LRRK2 contributes to PD risk is equivocal. Many assays, including direct assays of LRRK2 function, suggest that mutant alleles retain function. There is some direct human genetic evidence recently that loss of function alleles are present in humans but not pathogenic. If the authors wish to determine if pathogenic mutations associated with PD alter Mtb infection, they would have to use G2019S or R1441C knockin mice. Lastly, the phenotype of LRRK2 KO mice infected with Mtb has been previously reported (Hartlova et al) and the mild inflammatory phenotype seen by the present authors broadly resembles this previous report, though it is appreciated that there are some differences (e.g., no CFU or cytokine differences, unlike Hartlova). If the authors wish to 'correct the record', it might be appropriate to include the Mtb cfu/cytokine/neutrophil data (Figure S6A-C, 6A-B) as supplemental, along with a discussion of why you may have obtained different results (e.g., is this related to the use of +/- hets instead of +/+ WT mice?). However, we suggest 'saving' the neuroinflammation data for a subsequent paper when further work will allow some of the complex phenotypes to be better understood.

We agree and we already have LRRK2G2019S experiments ongoing in the lab for a subsequent publication. Instead of relegating the entire Mtb mouse infection to a supplementary figure, we have instead worked to expand this analysis and have added several panels to the figure (now Figure 8), These include:

· Mtb CFUs in *Lrrk2* KO and Het BMDMs (which show that Mtb replicates better in *Lrrk2* KOs at late infection time-point (Figure 8A)

· Additional readouts of neutrophil infiltration and inflammation in Mtb-infected lungs (Figure 8G-H)

· Gene expression changes in the lungs of Mtb infected mice (Figure, 8E-F), which shows lower levels of certain ISGs in *Lrrk2* KO mice vs. Het controls (which is consistent with our macrophage phenotype ex vivo)

We believe these in vivo data are important to highlight in the manuscript seeing as *LRRK2* is currently being targeted by drug companies for treatment of Parkinson’s Disease. Understanding the consequences of ablating *LRRK2* in the context of infection may help us understand potential side effects in patients taking *LRRK2* inhibitors (see final paragraph of the Discussion).

With regards to Härtlova et al., the takeaway from both of our Mtb infections is the same: there is more inflammation in the *Lrrk2* KO mice. We agree that our p-values may have suffered somewhat from our use of HETs. Our *Lrrk2* KO mice are also different. Ours, generated by the Michael J. Fox Foundation and repeatedly validated, has a KO of exons 39-40, which encode the kinase domain of the protein. Härtlova et al. use a strain that has a deletion of exon 2. Our Mtb strains are different as well. We use the highly virulent Erdman strain while they use H37Rv and N145. We are careful to point out potential explanations for discrepancies between our results the final paragraph of the Discussion.

2) The reviewers have significant concerns about how the data were analyzed. We feel it is inappropriate, e.g., in Figure 1 but also elsewhere, to present the response of control and KO genotypes to infection (normalized to unstimulated) separately from the baseline responses of genotypes at baseline. Typically, one would compare treatment and genotype in one experiment and use a two-factor analysis. Unless there is a strong rationale for performing the experiments separately and in fact that is what was done, this figure needs reworking and p values for genotype, treatment and interaction terms generated. We feel it is not appropriate to express results as fold induction when the baseline measurements are not equivalent across conditions. This concern also affects Figures 2, 3, 4E, 5G. The authors may have chosen their convoluted presentation because the basal levels of gene expression are very low compared to the induced levels and plotting all four conditions on the same axis would minimize the differences in basal expression. This might be solvable by plotting the data on a log scale; or if that is unsatisfactory, by plotting all the data in a single graph but with an additional inset to highlight the basal differences. In general, the statistical reporting should be improved by stating which test was used in which figure, what n precisely means in each figure and please also give the value of the test statistic, particularly for parent ANOVAs where these were used. Related to the above, in panels 3B, 3I, 3J, 4E, 5G and possibly elsewhere, please indicate if the 2nd and 4th conditions are statistically different from each other (i.e., does mitoT or ddC or cGAS KO, or urate, etc., significantly enhance the ability of LRRK2 KO cells to respond to stimuli?).

After numerous discussions with the editor and reviewers, we came up with an appropriate solution to these issues. All our data is presented as baseline (HET/WT vs. KO) separate from treatment (HET/WT vs. KO). No data in the revised manuscript is displayed as fold-change. In the majority of experiments involving induction or infection, the 2 (or 3) independent variables were heavily skewed (i.e. the variable of Mtb infection vs. variable of genotype) Therefore in order to avoid a type II error where genotype significance was 100% dependent on the strength of the stimulation, we performed a log transformation on the data prior to analysis. Data were then analyzed by two-way, or three-way ANOVA, followed by Tukey’s post-hoc test (see Materials and methods). We have also included the p-values for all parent and post-hoc tests in an additional Supplementary file. In order to depict baseline and induced gene expression on the same graph we have used either a log10 scale or broke the y-axis into segments where warranted. We are confident that the improved analysis will fully address the reviewers’ valid concerns.

For the RNAseq data in Figure 1: it was very hard to understand what the n of samples is. It appears from 1A and 1C that this an n=1 experiment, based on the lack of p values in A and the single lane in the heatmaps. If so, this is simply not an appropriate experiment to report. If not, then the presentation needs to show individual values for replicates.

Our original presentation of our RNA-seq data had our four replicates averaged. In response to reviewers’ concerns, we are now showing each of the 4 replicates separately. The updated heatmaps show gene expression differences between *Lrrk2* KO vs. Het at baseline (Figure 1A) and *Lrrk2* KO vs. Het at 4h post-Mtb infection (Figure 1C). The RT-qPCR validation of baseline and infection differences for several ISGs are now displayed on the same graph (Figure 1E) The new analysis is described in paragraph two of the Results section.

3) For Figure 4 and later in Figure 5, many studies have raised concerns about JC-1 because the red/green shift is dependent not just on membrane potential but also local concentration. Here, if the mitochondria show different volumes based on prior figures then this may contribute to some of the apparent effects. It is advised that JC-1 dye assays are run with CCCP controls to ensure the signal is sensitive to changes in mitochondrial membrane potential. Can the authors show this control data to ensure the signal changes are due to altered membrane potential. In addition, it is strongly recommended that these figures are confirmed with TMRM or TMRE, which are less prone to these artifacts.

We understand the caveats that come with JC-1 dye. We worked extensively with Thermo Fisher to decide on the proper dye for these types of experiments and they ultimately recommended JC-1 because it has a much larger dynamic range so it is generally more appropriate to assess phenotypes that may be more subtle. In any event, we did several major experiments with TMRE and observed the same trends in *Lrrk2* KO cells that we observed with JC-1 (Figure 4, 5, and Figure 5—figure supplement 1).We also added FCCP controls for JC-1 in Figure 5—figure supplement 1.

4) Can the authors demonstrate alterations in DRP1 phosphorylation by loss-of-Lrrk2 by western blot? The authors have shown images of DRP1 puncta, however there is no quantification to ensure no changes in overall DRP1 expression. Again, this should be done via western blot.

We did the western blot for phospho-S616 DRP1 and total DRP1 (Figure 4B, with a replicate blot in the Figure 4—figure supplement 1) and although the change is somewhat modest, it does confirm the flow results (Figure 4A) and demonstrates that there is no difference in total DRP1 protein levels between *Lrrk2* KD and control macrophages.

Furthermore the authors claim that DRP1 distribution is not different between genotypes but no quantification has been shown; by eye, it does indeed seem that there is decreased DRP1/TOM20 colocalisation in the Lrrk2-KO cells.

To address the point regarding DRP1/TOM20 colocalization, we’ve changed the language in the manuscript to make sure we are not overstating anything. We performed the IF to ask whether Drp1 distribution was grossly disrupted in *Lrrk2* KO macrophages. We can qualitatively observe that Drp1 localizes to the ends of mitochondrial tubules in both genotypes. Because the mitochondrial network is so fragmented in *Lrrk2* KO macrophages, there are perhaps more Drp1 dots because there are more pieces of network for it to be tethered to and yes, maybe less DRP1/TOM20 colocalization because the TOM20-positive pieces of the mitochondria are themselves smaller. Ultimately, we decided that there were too many differences in the mitochondrial networks themselves to allow quantification of DRP1 from these images. The image in question remains in Figure 4—figure supplement 1C and we do not make any claims about any quantifiable phenotypes.

As well, authors show histograms from flow data regarding DRP1-S616 levels, but this is not quantified anywhere.

All flow data in the main figures, including DRP1-S616, is now displayed as a histogram (Figure 4A, 4C, 4D, Figure 4—figure supplement 1C, Figure 7I-J)

Furthermore, the authors report an increase in DRP1 phosphorylation via flow in Lrrk2-KO BMDCs. This is in contrast to other reports that show increased LRRK2 kinase activity increases DRP1 (doi: 10.5607/en.2018.27.3.171). Can the authors speculate how both increased kinase LRRK2 activity and loss of LRRK2 can induce the same phenotype?

We have the same questions! We speculate that while the end result of increased DRP1 phosphorylation at S616 and mitochondrial fragmentation are shared between *Lrrk2* KO and *Lrrk2* G2019S cells, the molecular mechanisms driving these phenotypes are distinct. Our lab works closely with a colleague at TAMU, Dr. Phillip West, an expert in mitochondrial biology and together we have come up with several potential explanations. There are multiple important phosphorylated residues on DRP1 (T595, S616, S656, S637) and several kinases in addition to LRRK2 have been implicated in DRP1 phosphorylation (ERK2, PKA). It is possible that loss of LRRK2-dependent phosphorylation of DRP1 could allow these other kinases to activate it. Likewise, loss of LRRK2 could alter DRP1-containing protein complexes, which could result in altered DRP1 activity/phosphorylation that phenotypically mirrors overactivity of the LRRK2 kinase domain. We also like the idea that in one of the genotypes, (KO or G2019S) mitophagy and/or lysosomal dynamics are altered, and the accumulation of phosphoS616-DRP1 could be due to lack of mitochondrial turnover as opposed to the direct kinase activity of LRRK2. We are exploring these possibilities in ongoing experiments and have included some of this speculation in the Discussion section.